

**European UV DataBase (EUVDB) as a repository and**
**quality analyzer for solar spectral UV irradiance monitored**
**in Sodankylä**
**A. Heikkilä[1], J. Kaurola[2], K. Lakkala[1,3], J.M. Karhu[4], E. Kyrö[4], T. Koskela[1], O.**
**Engelsen[5], H. Slaper[6], G. Seckmeyer[7]**
[1]{Finnish Meteorological Institute, R&D / Climate Research, Helsinki, Finland}
[2]{Finnish Meteorological Institute, Weather and Safety, Helsinki, Finland}
[3]{Finnish Meteorological Institute, R&D / Arctic Research Center, Rovaniemi, Finland}
[4]{Finnish Meteorological Institute, R&D / Arctic Research Center, Sodankylä, Finland}
[5]{Norwegian Institute for Air Research, Tromsø, Norway}
[6]{National Institute for Public Health and the Environment, Utrecht, the Netherlands}
[7]{Leibnitz Universität Hannover, Institute of Meteorology und Climatology, Hannover,
Germany}
Correspondence to: A. Heikkilä (anu.heikkila@fmi.fi)
**Abstract**
Databases gathering atmospheric data have great potential not only as data storages but also in
serving as platforms for coherent quality assurance (QA). We report on the flagging system
and QA tools designed for and implemented in the European UV DataBase EUVDB
(http://uv.fmi.fi/uvdb/) for measured data on solar spectral UV irradiance. Spectra scanned by
the Brewer#037 MkII spectroradiometer in Sodankylä (67.37 °N, 26.63 °E) over the years
1990-2014 and uploaded into the database are examined using the inherent QA tools of the
database. The study demonstrates the performance of the QA tools of the EUVDB. In
addition, it yields an overall view of the availability and quality of the solar UV spectra
recorded in Sodankylä over a quarter of a century. Over 90 % of the four main quality
indicators are flagged as GREEN, indicating the highest achievable quality. For the BLACK





flags, denoting data not meeting the pre-defined requirements, the percentages for all the
indicators remain below 0.12 %.
**1    Introduction**
Monitoring the state of the Earth's atmosphere and the living conditions at the Earth's surface
requires measurements of high quality. This is a rule that applies to all atmospheric
parameters and variables, including solar UV irradiance. General guidelines for quality
control (QC) and quality assurance (QA) in solar UV irradiance measurements have been
carefully formulated (Webb et al. 1998, 2003; Seckmeyer et al. 2001, 2005, 2010). Following
the guidelines facilitates recognition of all potential sources of errors, reduction of the effect
of all those sources on the overall uncertainty of the measurements, and production of data of
as high quality as possible. However, to verify the efficiency of the recommended QC/QA
measures taken, tools to analyse the quality of data are needed.
Databases established for experimental atmospheric data have great potential, not only in
providing consistent formats, centralized collection, and efficient dissemination for large
amounts of data, but also coherent procedures for QA. The European UV DataBase EUVDB
(http://uv.fmi.fi/uvdb/) was established as a joint effort of 25 participants from 15 European
countries within the frame work of two projects: SUVDAMA (Scientific UV Data
Management) in 1996-1999 and EDUCE (European Database for UV Climatology and
Evaluation) in 1999-2002, funded by the $4^{th}$ and $5^{th}$ framework programmes of the EU,
respectively. The projects included development of comprehensive QA tools, to be applied to
all the spectral UV irradiance data submitted to the database.
The role of the centralized QA tools should not be seen as replacements for the existing on
site quality check procedures. Within the EUVDB, the data providers are therefore
encouraged to continue conducting their own QC actions and following the generally
accepted guidelines therein. The QA tools of the database are targeted to supplement the on-
site procedures, to verify that they have been applied successfully on the data, and give
uniformity to the data sets originating from a large number of geographically scattered





stations. Currently, the database includes 3 406 891 spectra, originating from 50 stations. The
database has got 111 registered users representing 61 different organizations. In addition, the
tools are meant to enable selection of data according to the requirements set by the objectives
of the planned study.
Over the operational years, the tools once developed have repeatedly proved to provide
valuable information on the quality of the data both to the data users and data providers.
Instruments measuring spectral UV irradiance are delicate and subject to several
environmental factors influencing their performance and the reliability of their measurements.
In addition, the maintenance of their calibration is challenging (e.g. Mäkelä et al. 2015).
Instrumental errors may lead to erroneous interpretations on the amount of UV radiation
reaching the surface of the Earth (e.g. McKenzie et al. 2015), and should be therefore
minimized in every possible way. Efforts to increase cost-effectiveness and automation in
ground-based measurements of solar UV irradiance further emphasize the importance of the
centralized QA tools.
We report on the tools implemented in the European UV DataBase for the quality assurance
of the solar spectral UV irradiance data uploaded by the data provider to the database. Data
measured by Brewer#037 MkII single monochromator spectroradiometer over the years 1990-
2014 in Sodankylä, Finland, are used to demonstrate the performance of the tools. The
selected data set provides a subject for a case study. The quality indicators are examined for
their frequency in general, and for selected case spectra in detail. For the first time, a detailed
comprehensive analysis on the quality of UV irradiance spectra measured by Brewer#037 in
Sodankylä is presented.
**2   Materials and methods**
**2.1   QA tools and flagging**
The QA performed on the UV irradiance data submitted to and stored in the EUVDB is based
on a package named CheckUVSpec containing two algorithms independent from each other:


AtmosphericSignature and ShicRIVM (http://www.rivm.nl/shicrivm; Slaper et al. 1995,
Williams et al. 2003).  The package produces indicators on the quality of the principal
elements of UV irradiance spectra: the wavelength scale, the irradiance scale, and the shape of
the spectrum.  In addition, it yields diagnostic information on atmospheric conditions and
variability of conditions during the scan. The package is an integrated part of the database,
run automatically in every event of data submission. Inclusion of several different indicators
allows checking of the data for different aspects.
The quality indicators are denoted as flags, associated with a selection of colours
(summarized in Table 1). GREEN flag is reserved for the spectra meeting the highest quality
criteria. YELLOW is used for those not fully complying with the highest criteria, but
satisfying the secondary criteria. RED flag is raised for a spectrum not meeting even the
secondary criteria, but still exceeding the rejection criteria. For spectra meeting the rejection
criteria, a BLACK flag is given. In case no definite conclusion on the quality of a doubtful
spectrum can be drawn, the spectrum is marked with a GREY flag. Flags are given for a
number of different properties of the spectrum. In addition, a master flag, describing the
overall quality, is given to the spectrum. The master flag is determined by the worst flag for
any of the quality indicators. For a GREEN master flag, all indicators have to be flagged as
GREEN. In case any of the indicators is BLACK, the master flag is BLACK.
AtmosphericSignature
The AtmosphericSignature QA tool is based on examination of differences between the
measured and modelled solar UV irradiance spectra. For model calculations of spectral UV
irradiance, the tool employs the FastRT program (http://zardoz.nilu.no/~olaeng/fastrt
/fastrt.html). Spectral UV irradiance is simulated for a range of well-defined atmospheric
scenarios, Description on the scenarios denoted as MIN, MAX, AERO, CLEA and CLOU are
given in Table 2. The tool examines the measured spectrum by comparison against the
scenarios. The closest match determines in which category of the scenarios the spectrum is
located. The scenario should comply with the actual conditions during the scan. If the




measurement conditions prove to differ from those of the scenario, the spectrum may be
considered suspect. The following cases of discrepancy are identified:
***Too high radiation level*:** The irradiance exceeds even the scenario of multiple scattering
including snow covered surface and clouds trapping and enhancing radiation on the ground.
***Enhanced radiation***: The irradiance exceeds the level normally encountered under clear sky
conditions. GREEN flag indicates highly reflective ground. In case of a YELLOW flag,
special cloud conditions must prevail in addition to the highly reflective surface.
***Clear sky*:** The irradiance indicates cloudless sky conditions.
***Moving clouds***: The irradiance spectrum contains features indicating clouds appearing or
disappearing during the scan.
***Clouds*:** The irradiance indicates cloudy conditions.
***Too low radiation level*:** The irradiance is even lower than that under extremely thick rainy
clouds.
***Too high solar zenith angle*:** The solar zenith angle during the scan exceeds 84°. The model
calculations do not yield results accurate enough. The flags are GREY.
The categories of the cases and the associated colours for the quality indicator Atm_signature
are summarized in Table 3.
ShicRIVM
ShicRIVM is a package developed for QA, correction and homogenisation of spectral UV
irradiance data. In the EUVDB, only the diagnostic (QA) part of the package is implemented
as an inherent quality analyser for the incoming data. No corrections on data are performed in
the database. The algorithm is able to detect shifts in the wavelength scale, determine the
lowest detectable irradiance level, and identify anomalies like spikes in the shape of the





spectrum. The flags are named as Shift1, Shift2, Start_irr, and Spike_shape. Descriptions on
the flags are given in the following.
***Shift1:*** Flag for detecting shifts in the wavelength range 300-325 nm. The shift in nm is given
in the detailed flag description of the spectrum. The colour of the flag is determined on the
basis of the following criteria: 0 nm < GREEN < 0.1 nm < YELLOW < 0.2 nm < RED < 0.4
nm < BLACK. In case the algorithm fails to yield at least five reliable shift determinations, or
if the general criteria indicate BLACK but the median irradiance in the vicinity of 310 nm
remains below 5e-4 W/m2/nm, the flag is marked as GREY.
***Shift2:*** Flag for detecting shifts in the wavelength range 325-400 nm. As the wavelength scale
of Brewer #037 ends at 325 nm, this flag is GREY for all Brewer #037 UV irradiance spectra
in the EUVDB.
***Start_irr:*** Flag for the lowest reliable irradiance reading. Five subsequent ratios of irradiance
readings are required to be within 25 % of the modelled ratios. Two numeric values are output
in the detailed flag description of the spectrum: irradiance at the first reliable reading, and the
highest irradiance below the first reliable reading. The colour of the flag is determined on the
basis of the following criteria set for the higher of the two irradiance values: GREEN < 5e-4
$W/m^2/nm$ < YELLOW < 1.5e-3 $W/m^2/nm$ < RED < 5e-3 $W/m^2/nm$ < BLACK. If the median
irradiance level around 310 nm is lower than 5e-4 $W/m^2/nm$, the flag is set as GREY
***Spike_shape***: Flag for spikes and anomalies in the local shape of the spectrum. If the ratio of
the irradiance reading and the median of 10 readings around the measured wavelength is over
two-fold as compared to that found in the spectrum obtained by model calculations, the
feature is interpreted as a spike, and a BLACK flag is returned. A RED flag results from a
ratio of two subsequent irradiance readings deviating more than 50 % from the corresponding
modelled ratio. In case the deviation exceeds 25 %, and yet remains below 50 %, a YELLOW
flag is given. The local shape is examined through the differences between the ratios of two
subsequent readings in the measured and the modelled spectrum. The following criteria are



used for the flagging of the local shape: GREEN < 10 % < YELLOW < 15 % <RED < 20 %
< BLACK. The worse of the two flags determines the overall colour of the flag Spike_shape.
Two indicators for the atmospheric conditions are included in addition to the flags mentioned
above. The first one is aimed at investigating the atmospheric transmission, and the second is
used to identify variability of conditions during the scan. The descriptions for these are given
in the following:
*Transmission_2*: Indicator for the average transmission in the wavelength range of 315-325
nm. The transmission is calculated by accounting for the Earth-Sun distance, and normalized
to unity for cloudless sky. Table 4 summarizes the descriptions on the defined categories of
transmission and the associated colours of the Transmission_2 flag.
*Scan_variability_2:* Indicator for the variability in the atmospheric transmission during the
scan for the wavelength range 325-400 nm. Diagnostic identifier for large variations
occurring during a scan. As the wavelength scale of Brewer #037 ends at 325 nm, this flag is
GREY for all Brewer #037 UV irradiance spectra in the EUVDB.
**2.2   Retrieval of flags for Sodankylä Brewer #037**
The quality of the solar UV irradiance spectra measured by the Brewer #037 in the Arctic
Research Center of the Finnish Meteorological Institute in Sodankylä (67.37 °N, 26.63 °E,
170 m a.s.l.) and submitted to the EUVDB database for the years 1990-2014 were examined
by retrieving the flag information concerning each individual spectrum. The retrieval was
done by using the PL/SQL tools provided by the database through its www-interface (Fig.1).
The interface allows the user to retrieve information on master flags of all colours or to
restrict the query on a subset of master flag colours. In addition, the user may retrieve
information on spectra measured in all atmospheric conditions, or concentrate on specific



prevailing conditions. In this study, the retrieval was done by choosing all colours of the
master flag and all kinds of atmospheric scenarios.
The files obtained by retrieval and containing the flags were examined by using an ad-hoc
script written in Perl (Practical Extraction and Report Language). Total number of each flag
colour for the indicators Shift1, Shift2, Start_irr, Spike_shape, Transmission_2 and
Atm_signature, and for the overall indicator Master, were computed in total and for each year
separately.
In addition to the calculations, the data were viewed through the tabular and graphical
summaries provided by the database itself. Figure 2 presents the monthly amount of UV
irradiance spectra measured by Brewer #037 over the years 1990-2014 as displayed on the
screen by the www-based interface to the database.
**3    Results and discussion**
The statistical calculations performed with the Perl script on the flag data retrieved for the
Sodankylä Brewer #037 from the EUVDB were examined in detail and summarized.
According to the calculations, the total number of spectra was found to vary between 4656
and 6724, except for the year 2011 for which only 876 spectra were found in the database.
The same phenomenon could be seen in the tabular and graphical summary provided by the
database interface. In Fig. 2, the monthly amounts of spectra over the year 2011 differ from
those of the other years.
Table 5 summarizes the statistics on the quality indicators as flag colours given to the
Sodankylä Brewer #037 UV irradiance spectra in the EUVDB. In terms of the indicator
Start_irr, over 99 % of the spectra were flagged as GREEN. For the indicator Transmission_2,
the percentage of the GREEN flags is over 95 %. For the indicators Shift1 and Spike_shape,
the relative frequencies of the GREEN flags are over 90 %. The percentage of the GREEN
flags for the indicator Atm_signature is only ~67 %, appearing relatively low in comparison.



The result is mainly explained by the fact that the algorithm of the tool AtmosphericSignature
sets the Atm_signature indicator to GREY whenever the solar zenith angle has been too high
(>84°) for accurate enough model calculations. This is a frequently occurring situation for
Sodankylä locating beyond the polar circle where the Sun might be low over several
consecutive scans after the sunrise and before the sunset.
According to the results listed in Table 5, the master flag given to the spectra is GREEN in
only about 61 % of the cases. This is because the master flag GREEN requires that all the
individual flags are GREEN. The relatively large fraction of the GREY flags for the indicator
Atm_signature is the major reason for the low amount of GREEN master flags. It is important
to note that this is due to the incapability of the radiative transfer model used by the QA tool
of simulating the UV irradiance spectrum needed as a reference to the measured one. From
the perspective of QA, more emphasis should be therefore put onto the other indicators.
However, the flag given to the Atm_signature indicator does yield useful information on the
prevailing atmospheric conditions. For Sodankylä Brewer #037, the Atm_signature is
therefore most applicable and in efficient use when cases of particular circumstances should
be extracted from the data set.
To demonstrate the performance of the QA tools and the flagging system of the EUVDB in
more detail, five cases were selected to be investigated more thoroughly. The case spectra and
the related flag information are summarized in Table 6. The corresponding spectra are also
shown in Figs. 3-7. In the following, each of the cases is examined separately.
**Case 1: All flags GREEN** (Fig. 3)
This is a typical summertime UV irradiance spectrum near the local noon, with cloudless or
almost cloudless sky. All quality indicators are GREEN, except Shift2, which is GREY for all
Sodankylä spectra, and Scan_variability_2, which has a value of NOT_DETERMINED, since
the wavelength range does not exceed 325 nm. No anomalies occur in the shape of the
spectrum.
**Case 2: Spike_shape flag BLACK** (Fig. 4)
This is a typical example of a spectrum containing a spike. In this spectrum, the spike occurs
as a sharp dip at wavelengths 315.5 nm and 316.0 nm. The reason for the spike is likely of
instrumental origin.
**Case 3: Start_irr flag BLACK** (Fig. 5)
This spectrum represents a case where the first reliable irradiance reading in the scan is
encountered at a wavelength too far in the scale. The algorithm has distinguished the first
reliable reading 0.034 W/m2/nm as far as at 319.5 nm.
**Case 4: Shift1 flag GREY** (Fig. 6)
No BLACK flags were given to the Shift1 indicator, and hence we chose to examine one of
the GREY flag cases here. This is a case where irradiance at 310 nm has remained below 5e-4
W/m2/nm due to low signal late in the evening and in the presence of clouds. The algorithm
cannot make any conclusions concerning possible shifts in the wavelength scale, and the
GREY flag is returned.
**Case 5: Transmission_2 flag BLACK** (Fig. 7)
Only two spectra were flagged in the studied data set as BLACK for the quality indicator
Transmission_2. In the case shown here, the ShicRIVM tool had associated the spectrum with
an atmosphere of extremely high (~2.3) transmission. The scan has been started at solar zenith
angle of ~85.5, and hence the AtmosphericSignature tool has not been able to make any
supporting conclusions on the atmospheric conditions. For the preceding scans, enhanced
irradiance has been noted by the AtmosphericSignature tool.
Analysis on the statistics of the flag information is obviously an efficient way to get an overall
view on different aspects of the data quality. However, the detailed examination of the
selected cases as described above gives a more profound insight into the function and
performance of the QA tools implemented in the database. Specifically, an understanding on
the metrics and categorization used by the different quality indicators helps the data provider
and the user in analysing and using the data in a meaningful way. Clearly, the indicators
provide an added value to the data set.



The potential of the indicators related to atmospheric conditions has not yet been fully
exploited in studies on the Brewer #037 UV irradiance data. Indeed, they could be used, for
instance, to screen cases of extreme conditions, to be used in studies focusing on some
particular atmospheric (radiative transfer) processes.
It should be noted that re-evaluation of the spectral UV irradiance data sets subjected to even
the most rigorous QC/QA procedures may reveal previously undetected features in the data
(see, e.g., Garane et al. 2006). This should not be seen as an invalidation of the QC/QA
procedures followed in the past. On the contrary, they should be seen as the necessary steps
having brought the data set to such a state that the discovery of the new features becomes
possible.
**4   Conclusions**
Solar spectral UV irradiance data measured in Sodankylä by Brewer #037 spectroradiometer
over the years 1990-2014 were studied through the repository features and the QA tools
provided by the European UV Database (EUVDB). The summaries on the data give an
overview on a consistent dataset extending over quarter of a century with only minor gaps.
The gaps found in the time series could be primarily traced to lamp measurements required
for the maintenance of calibration, and intercomparison campaigns that the instrument had
been participating, thus not being in operation by the home site at the time.
More importantly, the QA tools designed for and implemented in the database yielded
important information on the quality of the measured spectra. Only 0.12 % were flagged as
BLACK indicating severe flaws in the data. Over 90 % of the four main quality indicators are
flagged as GREEN, indicating the highest achievable quality.
For the master flag denoting the overall quality of the data, approximately 23 % were flagged
as GREY, denoting data that the QA tools were not able to make definitive conclusions on. A
majority of the cases of this kind could be traced to the indicator related to atmospheric
transmission and cases of high solar zenith angle (sza). Due to the restrictions in the
performance of the radiative transfer model FastRT used by the AtmosphericSignature tool,
the algorithm marks cases of sza greater than 84° all GREY. Therefore, for the Sodankylä



Brewer #037 UV irradiance spectra, the Master flag, and the relative amount of GREEN flags
given to the Master flag, are not the most relevant indicator for the overall quality. For that
purpose, the flags received by the individual quality indicators should be examined instead.
**Author contribution**
A. Heikkilä contributed to the design of the European UV DataBase (EUVDB) and the
implementation of the retrieval tools, conducted the statistical analysis for the Sodankylä case
study, and prepared the manuscript. J. Kaurola contributed to the design of the WWW
interface to the database and the tools implemented therein. K. Lakkala processed the spectral
UV irradiance data from Brewer #037 submitted to the EUVDB and performed the on-site
QC/QA actions on the data. J.M. Karhu made the lamp measurements required for the QC and
maintenance of the calibration of Brewer #037. E. Kyrö initiated the measurements of spectral
UV irradiance with Brewer #037 in 1989 in Sodankylä. T. Koskela contributed to the overall
QC/QA of the Brewer #037. O. Engelsen developed the AtmosphericSignature QA tool
implemented in the database. H. Slaper developed the ShicRIVM QA tool also implemented
in the database. G. Seckmeyer coordinated the projects SUVDAMA and EDUCE during
which the database and its QA tools were designed and implemented.
**Acknowledgements**
The authors gratefully acknowledge the financial support from the European Union granted
through the programmes FP4-ENV 2C and FP5-EESD for the projects SUVDAMA
(Scientific UV DAta MAnagement) and EDUCE (European UV Database for Ultraviolet
Radiation Climatology and Evaluation).



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

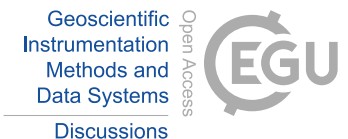

1    Table 1. General descriptions for the quality flags.

| Flag | Description |
| --- | --- |
| GREEN | Meets predefined* standard quality |
| YELLOW | Some (minor) deviation |
| RED | Indicates errors potentially problematic for certain applications |
| GREY | Flagging not possible |
| BLACK | Does not meet quality requirements |

2    *Criteria as recommended for type S-1 spectral instruments by Seckmeyer et al. 2001



1    Table 2. Atmospheric scenarious defined in and used by the QA tool AtmosphericSignature.

| Scenario | Description |
|---|---|
| MIN | The lowest naturally observable radiation levels, assumed to prevail under a homogenous, extremely thick water cloud (alto-stratus; Shettle 1990) with thickness of 4 km and liquid water column of 4000 $g/m^2$, equivalent to a cloud optical depth of 650 at a wavelength of 360 nm. No surface reflection. |
| MAX | Broken clouds scenario, the downwelling radiation assumed to be transmitted through a clear atmosphere and getting trapped between a snow-covered ground and an extremely thick homogenous alto-stratus cloud. The MAX scenario is assumed to yield the highest radiation levels obtainable naturally. |
| MAX_0 | The MAX scenario with all diffuse downward radiation transmitted through the clear atmosphere absorbed and scattered by the thick homogeneous cloud. |
| AERO | Cloudless but very turbid aerosol-loaded atmosphere with a visibility of 5 km. No surface reflection. |
| CLEA | Clear atmosphere with no aerosols or clouds present. No surface reflection. |
| CLOU | Cloudy atmosphere similar to MAX but with variable cloud density. Model simulations are run iteratively to find the cloud liquid water column yielding the best match between measurements and modelling. |



1    Table 3. Categories of different cases of discrepancies between the atmospheric scenarios

2    used by the QA tool AtmosphericSignature and the actual measurement conditions.

3    Associated colours for the Atm_signature flag.

| Category of cases | Atm_signature flag colour |
|---|---|
| Too high radiation level | BLACK or RED |
| Enhanced radiation | YELLOW or GREEN |
| Clear sky | GREEN |
| Moving clouds | GREEN |
| Clouds | GREEN |
| Too low radiation level | RED |
| Too high (>84 degrees) solar zenith angle | GREY |



1    Table 4. ShicRIVM categories of atmospheric transmission normalized to unity for cloudless

2    sky, and the associated colours for the Transmission_2 flag.

| Transmission | Transmission_2 flag colour | Description |
|---|---|---|
| >2.0 | BLACK | Extremely high transmission |
| >1.5 | RED | Very very high transmission |
| >1.25 | YELLOW | Very high transmission |
| >0.75 | GREEN | Low or no clouds |
| >0.25 | GREEN | Clouds |
| >0.10 | GREEN | Thick clouds |
| >0.05 | YELLOW | Very thick clouds |
| >0.01 | RED | Very very thick clouds |
| <0.01 | BLACK | Extremely low transmission |



1  Table 5. Absolute and relative frequencies of different flag colours given by the QA tools of
2  the European UV DataBase (EUVDB) for the UV irradiance spectra measured by the
3  Sodankylä Brewer #037 over the years 1990-2014.

| Flag colour | Shift1 | Start_irr | Spike_shape | Transmission_2 | Atm_signature | Master |
|---|---|---|---|---|---|---|
| GREEN | 117253 | 128580 | 116786 | 122793 | 86439 | 78674 |
| YELLOW | 0 | 129 | 10400 | 5461 | 14161 | 19524 |
| RED | 0 | 70 | 1467 | 479 | 63 | 521 |
| GREY | 11649 | 51 | 116 | 167 | 28239 | 30028 |
| BLACK | 0 | 72 | 133 | 2 | 0 | 155 |
| GREEN | 90.96 % | 99.75 % | 90.60 % | 95.26 % | 67.06 % | 61.03 % |
| YELLOW | 0.00 % | 0.10 % | 8.07 % | 4.24 % | 10.99 % | 15.15 % |
| RED | 0.00 % | 0.05 % | 1.14 % | 0.37 % | 0.05 % | 0.40 % |
| GREY | 9.04 % | 0.04 % | 0.09 % | 0.13 % | 21.91 % | 23.30 % |
| BLACK | 0.00 % | 0.06 % | 0.10 % | 0.00 % | 0.00 % | 0.12 % |

Total number of spectra: 128902



Table 6. Cases of UV irradiance spectra retrieved from the EUVDB as representatives for
different flag colours of the various quality indicators. Case no 1 used as a reference with all
flags GREEN; Cases 2-5 representing spectra for which a particular quality indicator flagged
as BLACK or GREY (in bold-type).

| No | Date | UTC | Shift1 | Start_irr | Spike_shape | Transmission_2 | Atm_signature | Master |
|----|------|-----|--------|-----------|-------------|----------------|---------------|--------|
| 1 | 21Jun03 | 10.01 | GREEN | GREEN | GREEN | GREEN | GREEN | GREEN |
| 2 | 8Jul03 | 7.45 | GREY | GREEN | **BLACK** | GREEN | GREEN | BLACK |
| 3 | 26Mar03 | 11.91 | GREY | **BLACK** | YELLOW | GREEN | GREEN | BLACK |
| 4 | 21Jun03 | 20.44 | **GREY** | GREEN | GREEN | GREEN | GREY | GREY |
| 5 | 9Oct92 | 14.14 | GREEN | GREEN | GREEN | **BLACK** | GREY | BLACK |



**Figure captions**
Figure 1. Screenshot on the www-based interface for making query into the EUVDB.  The
query may be restricted to a desired subset of flag colours.
Figure 2. Graphical summary on the spectral UV irradiance data measured by Brewer #037 in
Sodankylä in 1990-2014: Monthly amount of spectra as a screenshot of the on-the-screen
display by the www-based interface to the database.
Figure 3. Exemplar spectrum in case of a GREEN flag given to all individual indicators.
Spectrum measured by Brewer #037 in Sodankylä on Jun 21, 2003, 10.006 UTC.
Figure 4. Exemplar spectrum is case of a BLACK flag given to the shape of the spectrum
(indicator named *Shape_spike*). A dip in the spectrum at 316 nm is clearly distinguishable.
Spectrum measured by Brewer #037 in Sodankylä on Jul 8, 2003, at 7.4498 UTC.
Figure 5. Exemplar spectrum is case of a BLACK flag given to the lowest reliable irradiance
reading (indicator named *Start_irr*). Spectrum measured by Brewer #037 in Sodankylä on
Mar 26,  at 11.9067 UTC.
Figure 6. Exemplar spectrum in case of a GREY flag given to the potential shifts in the
wavelength scale (indicator named *Shift1*). Spectrum measured by Brewer #037 in Sodankylä
on Jun 21, 2003, 20.4425 UTC.
Figure 7. Exemplar spectrum in case of a BLACK flag given to the atmospheric transmission
(indicator named *Transmission_2*). Spectrum measured by Brewer #037 in Sodankylä on Oct
9, 1992, 14.136 UTC.





**Spectrum type and quality flag**

Make a choice of spectrum type and the quality flags for the spectra to be retrieved. For most of the instruments only spectrum type "global" is available. Quality flag refers to the master flag for wavelength scale errors, spectral shape errors and irradiance scale errors.

**Spectrum Type:** Global

| **Quality Flags (master flag)** |
| --- |
| ☑ Green |
| ☑ Yellow |
| ☑ Red |
| ☑ Grey |
| ☑ Black |

**Prevailing atmospheric condition**

Make a choice of prevailing atmospheric condition for the spectra to be retrieved according to CheckUVSpec tool by NILU and ShicRIVM tool by RIVM.

| **CheckUVSpec:** see infobox for details |
| --- |
| ☑ Too high radiation level (RED or BLACK) |
| ☑ Enhanced radiation(YELLOW or GREEN) |
| ☑ Clear sky (GREEN) |
| ☑ Moving clouds (GREEN) |
| ☑ Clouds (GREEN) |
| ☑ Too low radiation level (RED) |
| ☑ Too high solar zenith angle (>84 degrees, GREY)) |

| **ShicRIVM:** see infobox for details |
| --- |
| ☑ Extremely high (>2.0, BLACK) |
| ☑ Very very high (>1.5, RED) |
| ☑ Very high (>1.25, YELLOW) |
| ☑ Low or no clouds (>0.75, GREEN) |
| ☑ Clouds (>0.25, GREEN) |
| ☑ Thick clouds (>0.1, GREEN) |
| ☑ Very thick clouds (>0.05. YELLOW) |
| ☑ Very very thick clouds (>0.01, RED) |
| ☑ Extremely low (<0.01, BLACK) |





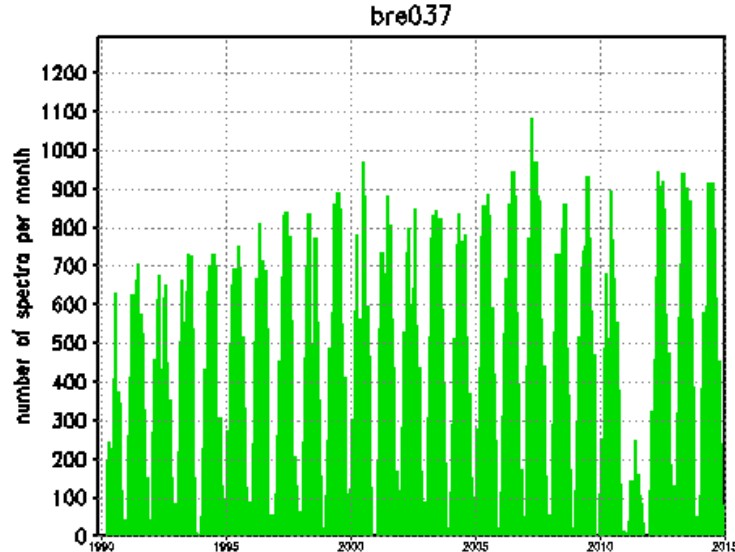





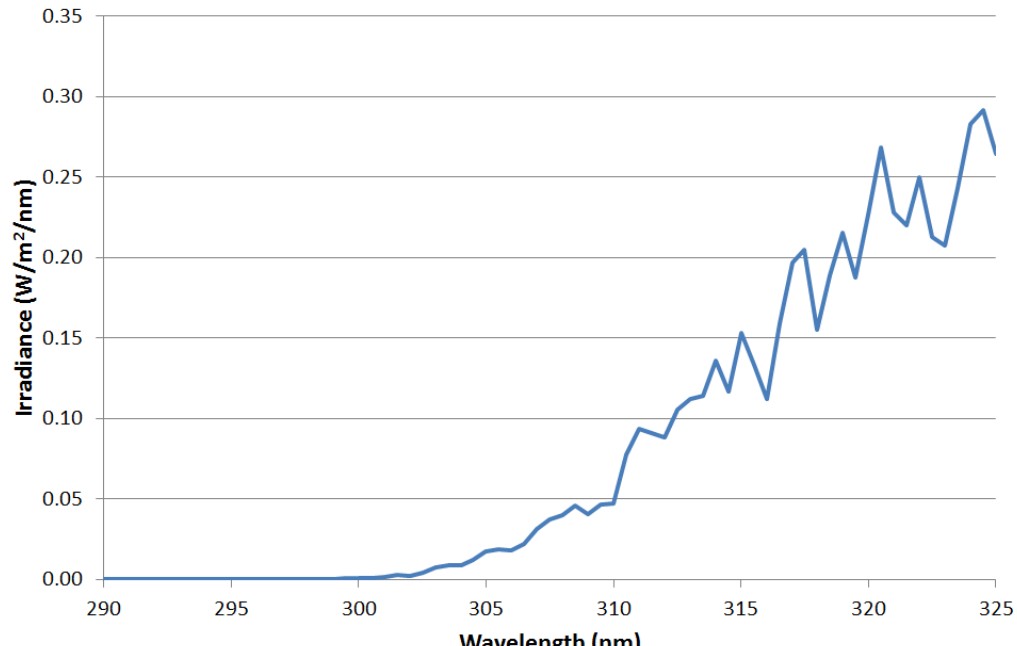





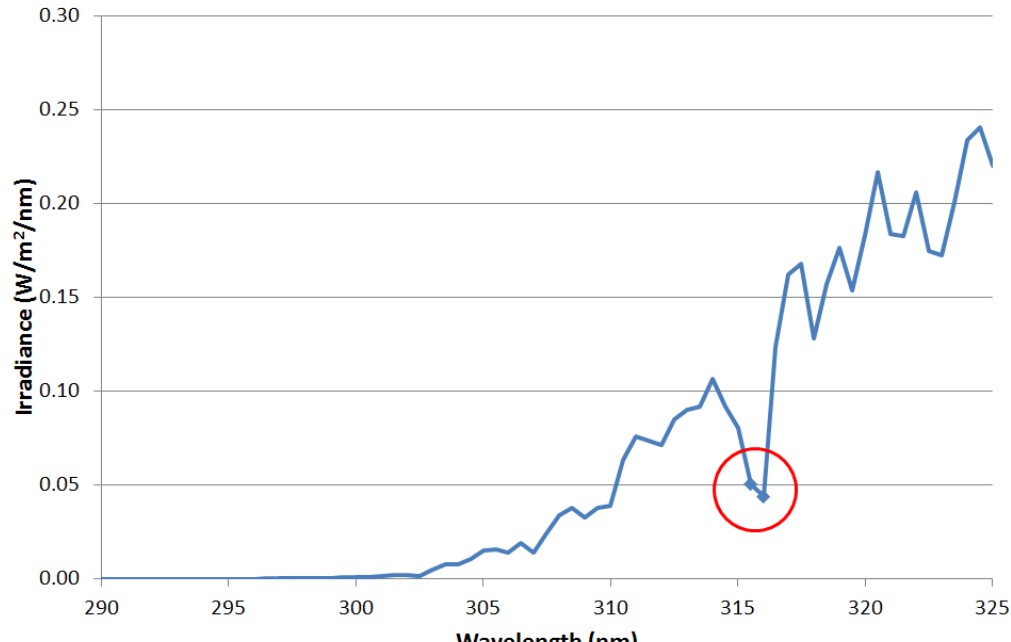




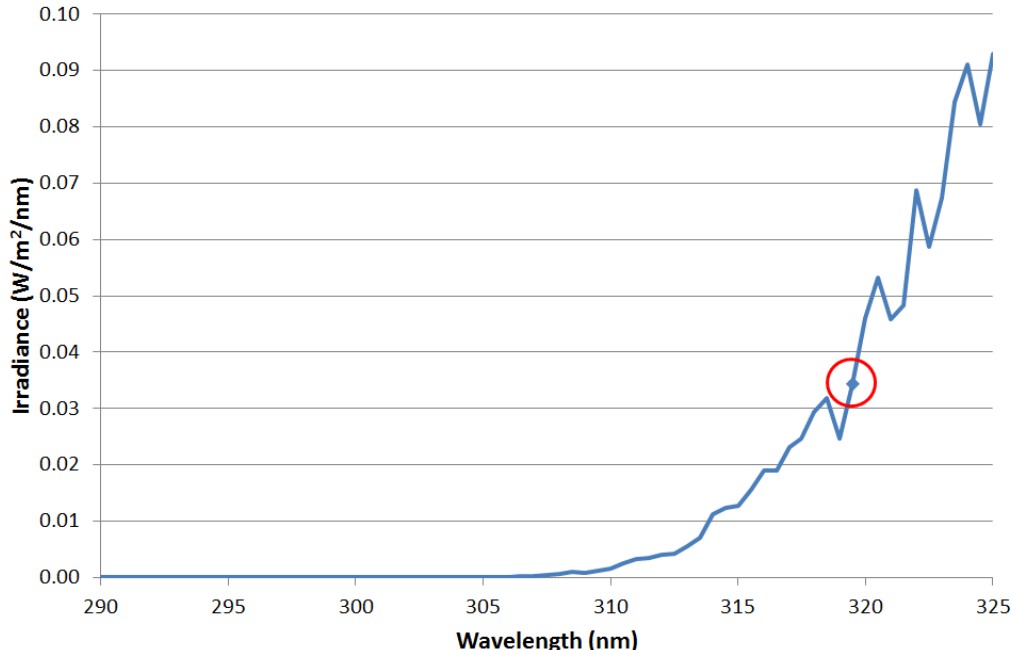





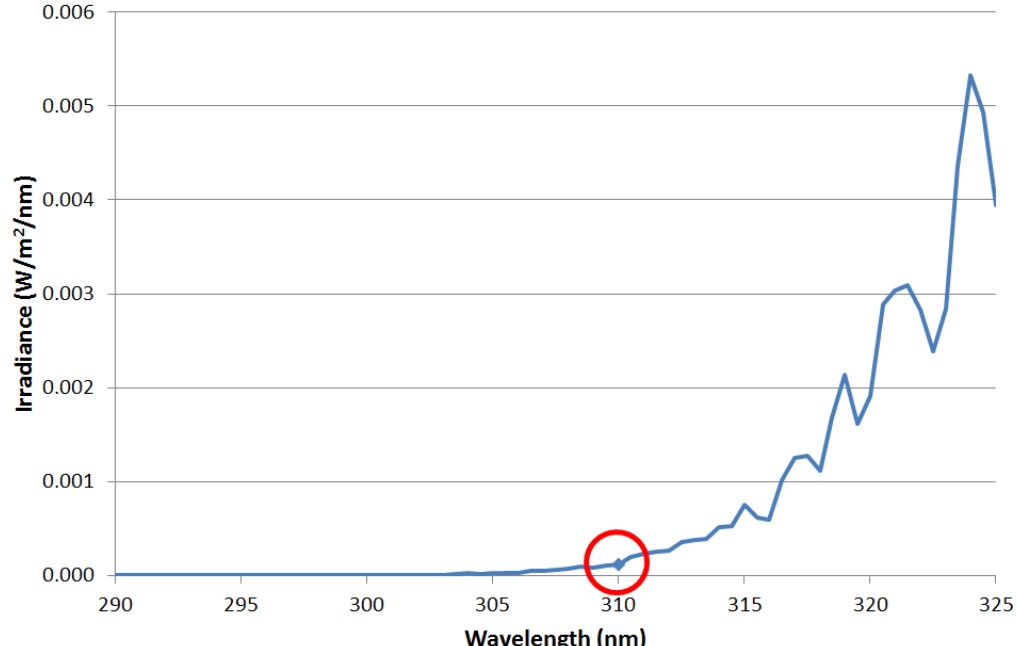





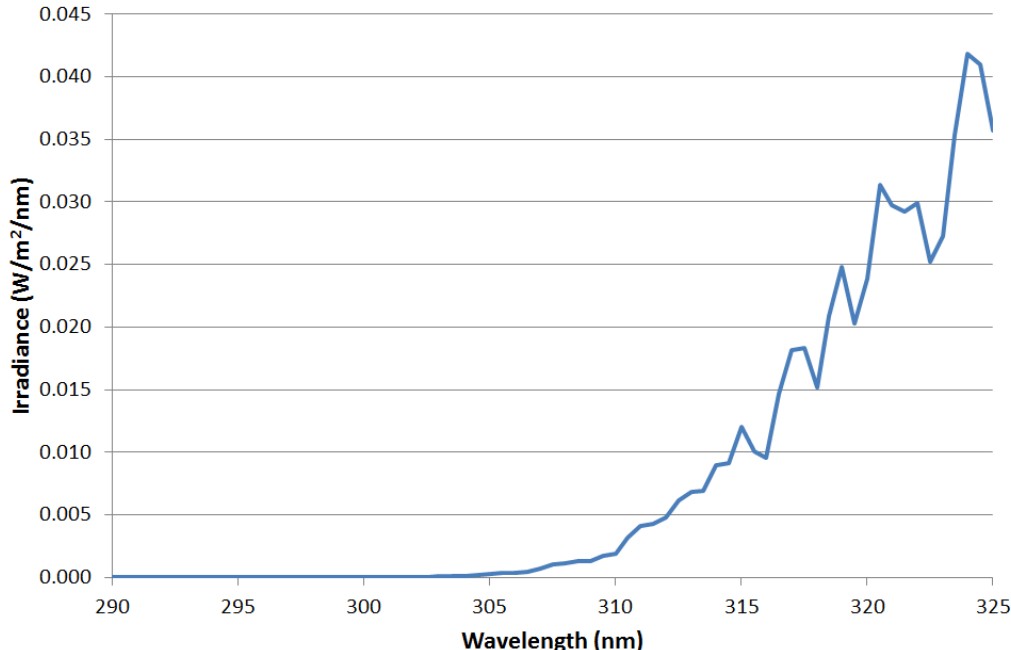