# Peer review of "European UV DataBase (EUVDB) as a repository and"

_Geoscientific Instrumentation, Methods and Data Systems, 2015_

## Referee Comment (RC1) · Anonymous Referee #1 · 4 Feb 2016

This paper details the QA aspect of the EUVDB, and the outcome of applying that QA to 25 years of UV data from Sodankyla. Neither the EUVDB, the QA tools, nor the Brewer spectrophotometer and its data are new, but the paper fits the remit of the journal by describing them as the focus of the manuscript rather than an adjunct to analyzing the data.

The manuscript aims to describe the QA system employed at the EUVDB, and then both demonstrate application of the QA tools and use them to assess the performance of the Brewer spectrophotometer measurements of UV data at Sodankyla over the past 25 years. This dual purpose application to Sodankyla data is in danger of becoming a circular argument, lacking as it does a critical assessment of whether the QA tools are

valid. There is some addressal of these issues in the case studies, but in the context of explaining away grey flags, rather than as a discussion of the use and validity of the QA system.In trying to do two things the manuscript does not quite succeed in doing either properly. This should be addressed before publication can be considered – some guidance is given below.

Top of P3 It would be helpful to have some further detail of the EUVDB repository e.g. to mention that data comes from both long-term monitoring sites and also campaign data. How many data records/stations are current (still regularly submitting data), and how many substantial data records exist (eg more than 10 years of data). What is the geographical extent of the submitting stations?

P3 line 7 – are data submitters actively alerted if there is a QA issue with their data, or does the system rely on the submitters checking for the QA flags? Can you prove this statement (line 6-7) eg by referencing publications that have used data from EUVDB.

P4 line 30 – how can the QA system determine whether the model scenario that the measurement matches was indeed the scenario under which the measurement was taken? It should be (as stated) but this cannot be determined by the software, so the quality becomes determined by whether the spectrum is "normal" i.e. meets expectation for the majority of times and places in Europe. This should not determine quality, as well illustrated by Sodankyla where many spectra are grey because of low SZA, not necessarily because the data are unrepresentative of the true conditions. See also comment on Figure 1.

P6 Shift1: Although the original description of shicRIVM is cited, a little more detail is needed to assist the reader of this manuscript e.g. the shift is assessed relative to what?

P6 Shift2: The description of this flag should not be discarded just because it does not apply to Brewers. The manuscript claims to describe the QA system of EUVDB, so it should describe it fully and completely, not as applied to a subset of the data.

P6 Start_irr: this description is very confusing. "Five subsequent ratios of irradiance readings..." What are the ratios (i.e. if a:b what are a and b?) and subsequent to what? What is the model to which this assessment is also applied? Which of these 5 ratios determines the first reliable reading (first?, fifth?), and why is a highest value below (assuming that below here means at a shorter wavelength) the first reliable reading potentially used to set the flag when it is by definition not reliable?

P6 Spike_shape: again a clearer description is needed e.g. (line 24) "the spectral irradiance reading at the measured wavelength and the median of 10 readings around the measured wavelength is over twice that of the matching model calculation...." As with Start-irr, the model should be explained somewhere before this point. Is it from AtmosphericSignature, or is it from within shicRIVM (as implied by Figure 1)? Line 30 "subsequent readings" could mean two scans one after the other. What I think you mean is two consecutive wavelengths in a single spectrum (measured or modeled). Please clarify. See also comment on Start_irr, which I think suffers from the same confusion.

P7 Scan-Variability_2: Please describe fully (see above comment on Shift2).

Figure 1 – this does not entirely agree with Table 6, nor with the description of the master flag in the text. The master flag in Figure 1 is stated as dependent on (taken as the worst of) wavelength errors, spectral shape errors (ie spikes) and irradiance scale errors (start irradiance). It does not include (according to Figure 1) the 2 versions of atmospheric transmission flag (from shicRIVM and AtmosphericSignature) that are included in table 6 as contributing to the master flag. Nor does it address the scan variability flag that has been ignored for the Brewer. Figure 1 is operational at the database and implies that a user can select a master flag that for the most part indicates instrument based quality, and then one or both (?) of two atmospheric condition indicators (that are not identical but very similar in their information). The manuscript should explain what a general user of the database can expect from the quality flags (as per Figure 1). If the Sodankyla data has combined the instrument master flag with

the atmospheric flags to give an overall flag then that should be explained separately. If not, then the page on the database for the user interface needs to be changed to be consistent with the applied meaning of master flag.

P8, line 19 .. the annual total number of spectra. . . Line 22 – why is there so little data in 2011. If this aspect of the manuscript now indicates the QA of the Sodankyla data we should be told.

P9 line 3 Suggest "This is a frequent occurrence for Sodankyla, located within the Polar circle, where the sun can be low for several consecutive scans after sunrise and before sunset."

Lines 7-17 Rather clumsily written. See also comment on figure 1 and develop the argument (eg should Atm_signature be part of the master flag?)

The combination of figures 3-7 should be explored. Figs 3 and 4 can definitely be combined, indeed are more instructive that way. Figure 5 might also be added. Alternatively Figure 5 could be combined with Figs 6&7.

The case studies are useful. It would also be helpful to show how selecting a certain flag would alter the data set eg select only master flag green and show how that influences the entire Sodankyla dataset – contrast to Fig 2.

P12 End of conclusion. The work done here has been performed and presented by those very familiar with the EUVDB and the Sodankyla Brewer in its unique setting. The last paragraph of the conclusion states that the master flag is not the most relevant overall, and more detailed exploration of flags (presumably aided by prior knowledge) is required. How would a novice user fare when trying to use the site and QA system. Could a comment on this be provided.

Minor points: P3 Line 4 What is "the planned study". Better just to say "according to the requirements of the user".

P4 line 7 aspects of what?

P4 line 27 grammar

Multiple cases of misuse of prepositions. These do not detract from the meaning but should be corrected in editing (one example contributes to the problems in line 27 above).

P10, Case study 3, and first paragraph of conclusion on P11 – rewrite (clumsy construction)

---

## Referee Comment (RC2) · Anonymous Referee #2 · 4 Mar 2016

Heikkilä et al., "European UV Database as a repository and quality analyzer for solar spectral UV irradiance monitored in Sodankylä"

The authors describe the quality assurance (QA) methodology that is currently used with the solar spectral irradiance measurements. Their approach comprises several metrics that provide important supplemental information about the actual spectral data. This metadata, or, in the authors' terminology, "flags", allows the end-users to assess the reliability of the data. For those actually carrying out the measurements using a spectroradiometer, the QA is an invaluable tool for instrument maintenance and calibration, which is crucial for research based on data covering several decades.

In my opinion, the manuscript is of relevance for the science community and suitable

for publication in Geoscientific Instrumentation, Methods and Data Systems. I do, however, have a few comments and recommend a minor revision before publication.

1. The definition of high quality is discussed in the introduction but only references to literature (Webb et al. and Seckmeyer et al.) are provided. In my opinion, the manuscript would benefit from having a short qualitative description of what actually is considered "standard quality".

2. Likewise, a brief description of the Brewer and its nominal operating mode(s) would be good to have. Perhaps the authors could also describe some of the routine operation challenges, if any, that can or could be effectively tackled by using the QA system rather than on-site routines.

3. (Results and discussion) Does the number of spectra (4656-6724) refer to the annual measurements? Why does this vary? Instrument trouble or do you only carry out measurements when certain criteria are met?

4. (Results and discussion, page 10, lines 22-28) The authors state that a detailed examination of the selected cases provides a more profound understanding of the function and performance of the QA methodology. While I agree that a closer look at the data does help in understanding why a certain flag is there, I don't think a small number of cases is sufficient for generalisation. Are you really sure that you would have arrived to the same conclusions if you had selected different spectra? Wouldn't it be much more useful to collect all spectra with, e.g., Shift1 GREY flag and analyse why the algorithm (built-in to the QA) cannot make any conclusions about wavelength scale shifts? Something like this would be an excellent topic for a follow-up study.

5. (Conclusions) Are the gaps in the time series not recorded in the EUVDB? Would it not be extremely useful for the end-users to quickly find out that there are no spectra for the time they are interested in?

6. (Conclusions) There were 23% of GREY flags for the overall quality. The authors

state that the majority of these indefinite conclusions could be traced to restrictions in the radiative transfer model FastRT that could not handle solar zenith angles above 84 degrees. Are there better models or has your quality flag analysis highlighted a gap in our knowledge? In both cases, these indefinite cases would probably be of high interested for modellers working on radiative transfer at higher latitudes.

7. (Table 2 and 3) Have you compared the cloudy flag with synoptic observations? Do they agree?

– Some minor comments:

8. (Abstract, page 1, lines 22-24): The sentence "Spectra scanned by..." is very complex. Could be simplified.

9. (Introduction, page 3, lines 21-22: I do not understand the sentence "The quality indicators are examined for their frequency in general..." Do you refer to "occurrence"?

---

## Author Comment (AC1) · 17 Jun 2016

Answer from the authors to the Interactive comment on "European UV DataBase (EU-VDB) as a repository and quality analyzer for solar spectral UV irradiance monitored in Sodankylä" by A. Heikkilä et al.

Anonymous Referee #1

The comments are answered below in the following sequential manner: Q denoting the original comment; A denoting the authors' answer to the comment, and C denoting the corrections and amendments to the manuscript.

[Figure]

Q: This paper details the QA aspect of the EUVDB, and the outcome of applying that QA to 25 years of UV data from Sodankyla. Neither the EUVDB, the QA tools, nor the Brewer spectrophotometer and its data are new, but the paper fits the remit of the journal by describing them as the focus of the manuscript rather than an adjunct to analyzing the data. The manuscript aims to describe the QA system employed at the EUVDB, and then both demonstrate application of the QA tools and use them to assess the performance of the Brewer spectrophotometer measurements of UV data at Sodankyla over the past 25 years. This dual purpose application to Sodankyla data is in danger of becoming a circular argument, lacking as it does a critical assessment of whether the QA tools are valid. There is some addressal of these issues in the case studies, but in the context of explaining away grey flags, rather than as a discussion of the use and validity of the QA system.In trying to do two things the manuscript does not quite succeed in doing either properly. This should be addressed before publication can be considered – some guidance is given below.

A: We can see the point presented by the Referee and realize both the expectations and challenges emerging from an impression that the study is aimed at evaluating the performance of the QA tools and the Brewer measurements at the same time. We have therefore carefully addressed the following guiding comments and made the corresponding changes in the manuscript. The demonstration of the tools is still restricted into the data measured by Brewer #037 in Sodankylä due to the scope of the issue (Special issue titled "Multi-disciplinary research and integrated monitoring at the Sodankylä research station: from sub-surface to upper atmosphere processes"). The analysis on the performance of the QA tools and the conclusion drawn are therefore strictly valid for this particular data set only. This is now more clearly stated in the revised manuscript. While preparing the revision, we have tried to clarify the scope of the study as the first examination into the performance of the tools through one selected case of a data set. We hope that we have succeeded in achieving this objective.

Q: Top of P3 It would be helpful to have some further detail of the EUVDB repository e.g. to mention that data comes from both long-term monitoring sites and also campaign data. How many data records/stations are current (still regularly submitting data), and how many substantial data records exist (eg more than 10 years of data). What is the geographical extent of the submitting stations?

A: We agree with the referee on this point. We have checked the type of data (spectral or broadband) submitted from each station, as well as the temporal coverage of each spectral UV irradiance data set from each site. The total number of spectra has increased to 3434610, the previous number from the time of writing of the manuscript having been 3 406 891 (checked on Dec 6, 2015). The increase is 27710 spectra in six months.

The exact total number of registered sites is currently 49. Of these, Briancon is registered twice. One station, Arenosillo, has been registered, but has submitted no data. The total number of different stations having submitted UV data is therefore 47.

Two of the sites registered in the database are known to be campaign sites: Military Airport Tatoi and Nea Mihaniona. The data submitted from these sites only extend over one or two months. In the database, no distinction between the campaign and permanent sites is made. However, all the other sites than Military Airport Tatoi and Nea Mihaniona have submitted data from clearly longer periods of times than just 1-2 months, in minimum for 6 months (Vindeln). They can be therefore considered sites with longer-term solar UV measurements.

The number of stations having submitted spectral UV data is 37, of which 2 campaign sites and 35 stations with long-term spectral UV measurements. There are hence 10 stations that have submitted broadband UV data, but no spectral UV data. Substantial records of solar spectral UV irradiance data extending over 10-23 years are available from 12 sites. Five of these sites provide data sets covering over 20 years of measurements. These sites are the very same that continue reporting their spectral UV data regularly to the database.

A majority of the sites having submitted data in the database locate in Europe. One site (Lauder) is in New Zealand, and one (Princess Elisabeth Antarctic base) in the Antarctica. From Lauder, spectral UV data has been submitted, whereas from Princess Elisabeth Antarctic base, broadband UV data has been reported. On the northern hemisphere, the geographical coverage of the database extends from 28.49 to 69.66 in latitude and from -26.6330 to 27.2233 in longitude. The Northernmost site is the Auroral Observatory in Tromso, Norway. The Southernmost site is Centro de Investigación Atmosférica de Izaña on the island of Tenerife, Spain. The Easternmost site is Sodankylä in Finland (long -26.6330) and the Westernmost Angre do Heroismo (long 27.2233) on one of the Azorean islands of Portugal. The altitudes of the sites vary from 2 m a.s.l. (Leba in Poland) to 3106 m a.s.l. (Sonnblick in Austria).

C: A new paragraph featuring the repository of EUVDB in more detail has been added in the chapter of Introduction. The following sentences:

"Currently, the database includes 3 406 891 spectra, originating from 50 stations. The database has got 111 registered users representing 61 different organizations. In addition, the tools are meant to enable selection of data according to the requirements set by the objectives of the planned study."

-> has been replaced by the following paragraph:

"Up to date, the database has got 111 registered users representing 61 different organizations. The number of registered stations is 49, of which 47 has submitted spectral and/or broadband data on solar UV irradiance over the years of the existence of the database. Two sites locate outside Europe: Lauder in New Zealand and Princess Elisabeth Antarctic base in the Antarctica. The number of stations having reported spectral UV data is 37, of which 35 stations with long-term spectral UV measurements and two sites having hosted a measurement campaign. On the northern hemisphere, the geographical coverage of the database extends from 28.49N (Izaña, Tenerife, Spain) to 69.66N (Tromsø, Norway) in latitude and from 27.22W (Angre do Heroismo, Azores,

Portugal) to 26.63E (Sodankylä, Finland) in longitude. The altitudes of the sites vary from 2 m a.s.l. (Leba, Poland) to 3106 m a.s.l. (Sonnblick, Austria). Substantial records of solar spectral UV irradiance data extending over 10-23 years are available from 12 sites. Five of these sites provide data sets covering over 20 years of measurements. These sites are the very same that continue reporting their spectral UV data regularly to the database."

Q: P3 line 7 – are data submitters actively alerted if there is a QA issue with their data, or does the system rely on the submitters checking for the QA flags? Can you prove this statement (line 6-7) eg by referencing publications that have used data from EUVDB.

A: In every submission event, an automatic report on the correctness of the file format and the success of the submission is immediately given to the submitter. Every file has to comply with the defined syntax developed within the SUVDAMA project and named flexstor. A program named flxcheck is implemented in the database to check the files for their syntax but also for their contents and the availability of the so called include files providing additional information concerning a large set of spectra. These data include characteristics of the measurement site (e.g., the horizon around the site) and the instrument (e.g., the slit function of the monochromator). On the basis of the results of the check, a status flag is given to each spectrum. Six different status flags are currently in use: normal, variant, unusal, suspect, mistaken, and corrupt. The files flagged as corrupt are rejected, all others are accepted. Nevertheless, the status flag is also added onto the first line of the submitted file, together with the date of the check. In addition, a complete error message stating the problem encountered in which file and at which line is returned to the user. The status flags are associated with the syntax of the data files submitted to the database, not the quality of the data themselves, and are not therefore dealt with in this study.

The QA flags are produced within the next phase of the upload process where the data in the accepted Flexstor formatted datafiles, stored in the Flexstor directory tree of the

database, are extracted and loaded into the tables in the Oracle relational database. The data submitters are not specifically alerted if their data have been tagged with QA flags of specific colors. Instead, it is indeed implicitly assumed that the submitters are interested in their own data in an extent that it urges them to retrieve the QA flags for their data. However, there are no tools up to date to investigate or statistics made on how many of the data providers do that. The QA tool CheckUVSpec and the ShicRIVM included in it are freely available. The data providers can therefore install them on their home computers and check their data with them before the submission. Some of the data submitters are known to do the check either using just ShicRIVM or the complete CheckUVSpec. Unfortunately, we have no extensive statistics on how many do such a check prior to submission.

The number of publications related to the EUVDB is substantial (e.g.: Seckmeyer 2002, Seckmeyer 2004, for the time periods of SUVDAMA and EDUCE projects, respectively). However, we have no separate statistics on the publications using either the spectral UV irradiance data or the related QA data retrieved from the EUVDB. According to the data protocol of the EUVDB, the user of the data is only obliged to inform the submitter on the use of his/her data and offer him/her co-authorship in any publication using the data. After the end of the EDUCE project, it has been therefore not possible for us to maintain a complete record on the publications using the data retrieved from the EUVDB.

References:

Seckmeyer, G. Publications within SUVDAMA (2002). Available at: http://www1.muk.uni-hannover.de/∼seckmeyer/EDUCE/results/suvrefs.html. Last modified Jul 15, 2002. Last accessed Jun 12, 2016.

Seckmeyer, G. Publications within EDUCE (2004). Available at: http://www1.muk.uni-hannover.de/∼seckmeyer/EDUCE/results/publications.html. Last modified Feb 26, 2004. Last accessed Jun 12, 2016.

C: We have rephrased the sentence pointed out by the Referee (on Page 3 Line 7) to avoid the potential impression that we have got quantified and complete information on the use of the QA tools of the database. The original sentence:

"Over the operational years, the tools once developed have repeatedly proved to provide valuable information on the quality of the data both to the data users and data."

now reads:

"Over the operational years, the tools once developed have provided valuable information on the quality of the data, readily available to both the users and the providers of the data (providers (Seckmeyer, 2002, 2004)."

To illustrate the usefulness of the QA tools provided by the database, we have added a paragraph in the chapter of "Results and discussion" giving references to a few representative studies that have made use of the QA tools. The paragraph reads as follows:

"In general, the required quality depends on the scientific question. These could be site specific issues or questions in a wider context, analyzing geographical differences and their causes, for example, as has been done by Seckmeyer et al. (2008a, 2008b). For these two studies, spectra with GREEN flags have been used only. Alternatively, the analysis may focus on a specific question like estimating probability functions (Voskrebenzev et al, 2015), where more spectra with non-GREEN flags may be included."

We have also added the following references into the list of References:

Seckmeyer, G. Publications within SUVDAMA (2002). Available at: http://www1.muk.uni-hannover.de/~seckmeyer/EDUCE/results/suvrefs.html. Last modified Jul 15, 2002. Last accessed Jun 12, 2016.

Seckmeyer, G. Publications within EDUCE (2004). Available at: http://www1.muk.uni-hannover.de/~seckmeyer/EDUCE/results/publications.html. Last modified Feb 26, 2004. Last accessed Jun 12, 2016.

Seckmeyer, G., Glandorf, M., Wichers, C., McKenzie, R., Henriques, D., Carvalho, F., Webb, A., Siani, A.-M., Bais, A., Kjeldstad, B., Brogniez, C., Werle, P., Koskela, T., Lakkala, K., Gröbner, J., Slaper, H., den Outer, P., & Feister, U. (2008a). Europe's darker atmosphere in the UV-B. Photochemical & Photobiological Sciences, 7(8), 925-930.

Seckmeyer, G., Pissulla, D., Glandorf, M., Henriques, D., Johnsen, B., Webb, A., Siani, A.-M., Bais, A., Kjeldstad, B., Brogniez, C., Lenoble, J., Gardiner, B., Kirsch, P., Koskela, T., Kaurola, J., Uhlmann, B., Slaper, H., den Outer, P., Janouch, M., Werle, P., Gröbner, J., Mayer, B., de la Casiniere, A., Simic, S., & Carvalho, F. (2008b). Variability of UV irradiance in Europe. Photochemistry and Photobiology, 84(1), 172-179.

Voskrebenzev, A., Riechelmann, S., Bais, A., Slaper, H., & Seckmeyer, G. (2015). Estimating probability distributions of solar irradiance. Theoretical and Applied Climatology, 119(3-4), 465-479.

Q: P4 line 30 – how can the QA system determine whether the model scenario that the measurement matches was indeed the scenario under which the measurement was taken? It should be (as stated) but this cannot be determined by the software, so the quality becomes determined by whether the spectrum is "normal" i.e. meets expectation for the majority of times and places in Europe. This should not determine quality, as well illustrated by Sodankyla where many spectra are grey because of low SZA, not necessarily because the data are unrepresentative of the true conditions. See also comment on Figure 1.

A: We agree with the Referee on the statement that the software cannot make an absolutely definite determination on whether the spectrum has been measured under the conditions suggested by the different scenarios. The choice of the category does possess uncertainty of its own, as do both the measurement and the model simulation of the solar UV irradiance spectrum. The definitions of the metrics aimed at assessing the compliance of the measurement with the different scenarios were subject to extensive

discussion and examination within the EDUCE project, and so were the treshold values for the metrics to be set for the different categories. The metrics and their thresholds could be and indeed would be a very interesting topic for a re-examination.

We realize that the reader may get an erroneous impression on the capability, or perhaps even the infallibility, of the categorization scheme. We have therefore revised the description on how the measured and simulated data are compared and the measured spectra categorized through the use of specific metrics. We have also added a description on the metrics to clarify how it is used to assess the potential measurement conditions into Table 3. In addition, we have included a note on how the thresholds for the metrics have been defined.

C: The last sentences in the paragraph describing the AtmosphericSignature tool have been rephrased, starting from line 28 on Page 4. The original sentences read as follows:

"The tool examines the measured spectrum by comparison against the scenarios. The closest match determines in which category of the scenarios the spectrum is located. The scenario should comply with the actual conditions during the scan. If the measurement conditions prove to differ from those of the scenario, the spectrum may be considered suspect. The following cases of discrepancy are identified:"

This has been now rephrased as follows:

"The irradiance of the measured spectrum is compared to that of the simulated scenarios on two narrow instrument specific wavelength bands, one on the lower end and one in the upper end of the measured spectral region. For the single monochromator Brewer #037 with a spectral range of 290-325 nm, only the lower band (at 315.0-316.0 nm) is used. The metrics used for the flagging is listed in Table 3. Examination of the metrics results in placing the spectrum into one of the following categories:"

We have also added the description on the metrics into Table 3. In addition, we have

added the following description of the two different cases in the description of the case "Moving clouds":

"Two different cases are dealt with: CL-NCU (Cloud at Lower band - No Cloud at Upper band), and NCL-CU (No Cloud at Lower band - Cloud at Upper band)"

The sentence after the categories of cases (Page 5 line 16) has been also changed. The original sentence reads as follows:

"The categories of the cases and the associated colours for the quality indicator Atm_signature are summarized in Table 3."

This has been replaced by the following sentences:

"The threshold values for the metrics given in Table 3 were subject to exhaustive discussion and set within the EDUCE project. In the QA tool running in the database, they are implemented as constant values and cannot be changed by the submitter nor the user of the data."

Q: P6 Shift1: Although the original description of shicRIVM is cited, a little more detail is needed to assist the reader of this manuscript e.g. the shift is assessed relative to what?

A: We agree that the reader would benefit from an extended description. We have therefore extended the description of the ShicRIVM tool and the quality indicator Shift1.

C: We have revised the description on ShicRIVM. The original description was as follows:

"ShicRIVM is a package developed for QA, correction and homogenisation of spectral UV irradiance data. In the EUVDB, only the diagnostic (QA) part of the package is implemented as an inherent quality analyser for the incoming data. No corrections on data are performed in the database. The algorithm is able to detect shifts in the wavelength scale, determine the lowest detectable irradiance level, and identify anomalies like spikes in the shape of the spectrum. The flags are named as Shift1, Shift2, Start_irr, and Spike_shape. Descriptions on the flags are given in the following."

The revised description reads as follows:

"ShicRIVM (www.rivm.nl/shic) is a software package developed for QA, correction and homogenisation of spectral UV irradiance data. It uses the Fraunhofer lines in the solar UV spectrum for the wavelength alignment of the measured ground-based spectrum. The structures have been derived from solar measurements at Kitt Peak National Observatory in Arizona, US (Kurucz et al. 1984), resulting in a high resolution extraterrestrial (ET) spectrum with a highly accurate wavelength scale. The ET spectrum is multiplied with the simulated atmospheric transmission and convolved with the instrumental slit function of the spectroradiometer. The atmospheric transmission is calculated using a modified version of the simple model of McKenzie (1991). The algorithm is not sensitive to the applied model for the atmospheric transmission, because it uses the local spectral structures (0.5-2 nm off the nominal) dominated by the Fraunhofer structures.

The software is able to detect shifts in the wavelength scale, determine the lowest detectable irradiance level, and identify anomalies like spikes in the shape of the spectrum. It has been extensively tested with a variety of spectrometers in various conditions during large intercomparison campaigns of spectroradiometers (e.g., CAMSSUM 1995 (Gardiner and Kirsch, 1997), SUSPEN 1997 (Bais et al., 2001)), NOGIC 1996 (Koskela et al., 1997), NOGIC 2000, MAUVE/CUVRA, QASUME 1999) and several smaller campaigns. The accuracy of the wavelength check has been shown to be less than 0.02 nm for spectrometers with FWHM less than 1 nm (Slaper and Koskela, 1996).

In the EUVDB, only the diagnostic (QA) part of the package is implemented as an inherent quality analyser for the incoming data. Consequently, no corrections on data are performed in the database. For the diagnostics, the tool uses specific indicators (flags) named Shift1, Shift2, Start_irr, and Spike_shape. Descriptions of the indicators

are given in the following."

We have also rephrased and extended the description of the quality indicator Shift1 to read as follows:

"Shift 1 The wavelength shift is calculated by using the ratio of irradiance at each wavelength to the irradiance at the neighbouring two wavelengths, effectively quantifying the fine structure of the spectrum. The ratios computed from the (ground-based) measured and modelled spectrum are compared. As the fine structures of the modelled and the measured spectra should be similar, a difference between the two ratios reveals a shift in the wavelength scale. The algorithm is described in detail in Slaper et al., 1995."

We have also added the following references in the list of references:

Bais, A.F., Gardiner, B., Slaper, H., Blumthaler, M., Bernhard, G., McKenzie, R., Webb, A.R., Seckmeyer, G., Kjeldstad, B., Koskela, T., Kirsch, P. J., Gröbner, J., Kerr, J.B., Kazadzis, S., Leszczynski, K., Wardle, D., Josefsson, W., Brogniez, C., Gillotay, D., Reinen, H., Weihs, P., Svenoe, T., Eriksen, P., Kuik, F. & A. Redondas (2001). SUSPEN intercomparison of ultraviolet spectroradiometers. Journal of Geophysical Research, 106(D12), 12509-12525.

Gardiner, B. G., and P. J. Kirsch, Intercomparison of ultraviolet spectroradiometers, Ispra, 24–25 May 1995, in Advances in Solar Ultraviolet Spectroradiometry, Air Pollut. Res. Rep. 63, pp. 67–151, edited by A. R. Webb, Eur. Commun., Luxembourg, 1997.

Koskela, T., Johnson, B., Bais, A. F., Josefsson, W., and Slaper, H.: Spectral sky measurements, in: Kjeldstad, B., Johnson, B., and Koskela, T.: The Nordic intercomparison of ultraviolet and total ozone instruments at Izana, October 1996, Izana, Meteorological Publications 36, Finnish Meteorological Institute, p. 109-139. 1997.

Kurucz, R. L., Furenlid, I., Brault, J., & Testerman, L. (1984). Solar flux atlas from 296 to 1300 nm. National Solar Observatory Atlas, Sunspot, New Mexico: National Solar Observatory, Harvard University Press, Cambridge, MA.

McKenzie, R. (1991). Application of a simple model to calculate latitudinal and hemispheric differences in ultraviolet radiation. Weather and Climate, 11, 3-14.

Slaper, H., and Koskela, T.: Methodology of intercomparing spectral sky measurements, correcting for wavelength shifts, slit function differences and defining a spectral referencce, in: Kjeldstad, B., Johnson, B., and Koskela, T.: The Nordic intercomparison of ultraviolet and total ozone instruments at Izana, October 1996, Izana, Meteorological Publications 36, Finnish Meteorological Institute, p. 89-108. 1997.

Slaper, H., Reinen, H., Blumthaler, M., Huber, M. & Kuik, F. 1995, "Comparing groundăĂŘlevel spectrally resolved solar UV measurements using various instruments: A technique resolving effects of wavelength shift and slit width", Geophysical Research Letters, vol. 22, no. 20, pp. 2721-2724.

Q: P6 Shift2: The description of this flag should not be discarded just because it does not apply to Brewers. The manuscript claims to describe the QA system of EUVDB, so it should describe it fully and completely, not as applied to a subset of the data.

A: We can see the need for the explanation of the quality indicator Shift2. Although the wavelength scale of the single monochromator Brewer like Brewer #037 does not extend to this wavelength, there are many spectroradiometers, Brewer double monochromators and others, that do extend up to this wavelength. We have therefore included the description of Shift2 in Chapter 2 (Materials and methds) in the sub section explaining the quality indicators of the ShicRIVM program.

C: We have completed the description of Shift2 in the manuscript as follows:

"Shift2: Flag for detecting shifts in the wavelength range 325-400 nm. As the wavelength scale of Brewer #037 ends at 325 nm, this flag is GREY for all Brewer #037 UV irradiance spectra in the EUVDB."

->

"Shift 2: Flag for detecting shifts in the wavelength range of 325-400 nm. The shift in

nm is given in the detailed flag description of the spectrum. The colour of the flag is determined on the basis of the following criteria: 0 nm < GREEN < 0.1 nm < YELLOW < 0.2 nm < RED < 0.4 nm < BLACK. In case the wavelength range of the spectrum does not extend over 325 nm, value 9.999 is returned and the indicator is flagged as GREY. If a wavelength shift is definitely detectable at less than five wavelengths, or if the flag is BLACK but the median irradiance around 310 nm is lower than 5e-4 Wm-2nm-1, the flag is returned as GREY."

Q: P6 Start_irr: this description is very confusing. "Five subsequent ratios of irradiance readings: : :" What are the ratios (i.e. if a:b what are a and b?) and subsequent to what? What is the model to which this assessment is also applied? Which of these 5 ratios determines the first reliable reading (first?, fifth?), and why is a highest value below (assuming that below here means at a shorter wavelength) the first reliable reading potentially used to set the flag when it is by definition not reliable?

A: We have tried to clarify the procedure and reasoning behind it. The spectral irradiance readings at the shortest wavelengths in a spectral scan are usually subject to a large uncertainty due to noise and occasional electronic spikes. This implies that the local shape of the spectrum at these short wavelengths does not resemble the expected shape according to the Fraunhofer and ozone absorption structures. This is checked iteratively on a series of five ratios of irradiance readings at subsequent wavelengths at the low end of the spectrum until the criterion is met. If below that first wavelength higher irradiance levels occur in the noisy range of the measured spectrum, the start irradiance level is increased to that level, since it obviously was too low.

It should be noted that the procedure does not strongly depend on the modelled spectrum since the Fraunhofer structures dominate the local shape of the spectrum (at 0.5 -1.5 nm steps). A simple radiative transfer model is used with a rough estimate on the ozone absorption based on the ozone structures in the irradiance measurement.

The procedure followed in ShicRIVM was tested against other methods, such as the

spectral ratio method (Bernhard et al. 1998) where the noise level is determined by dividing subsequently measured spectra from the same instrument. The results were comparable for a number of different instruments and spectra taken at a range of solar zenith angles.

Reference:

Bernhard, G., Seckmeyer, G., McKenzie, R., & Johnston, P. (1998). Ratio spectra as a quality control tool for solar spectral UV measurements. Journal of Geophysical Research: Atmospheres, 103(D22), 28855-28861.

C: We have revised and extended the description of the quality indicator Start_irr. The original description was as follows:

"Start_irr: Flag for the lowest reliable irradiance reading. Five subsequent ratios of irradiance readings are required to be within 25 % of the modelled ratios. Two numeric values are output in the detailed flag description of the spectrum: irradiance at the first reliable reading, and the highest irradiance below the first reliable reading. The colour of the flag is determined on the basis of the following criteria set for the higher of the two irradiance values: GREEN < 5e-4 W/m2/nm < YELLOW < 1.5e-3 W/m2/nm < RED < 5e-3 W/m2/nm < BLACK. If the median irradiance level around 310 nm is lower than 5e-4 W/m2/nm, the flag is set as GREY."

The description now reads:

"Start_irr: Flag for the lowest reliable irradiance reading. In order to determine where the reported spectral irradiance starts to exceed the noise level of the instrument, the algorithm takes the ratio of the measured spectral irradiance at the lowest reported wavelength to the spectral irradiance at the next higher wavelength. Five such ratios at subsequent wavelengths are compared with simulated modelled ratios. If all these ratios compare within 25% with the modelled ratios, the lowest of these readings is taken as the irradiance level at which the spectrum becomes reliable. If the criterion is

not met, the same procedure is repeated starting with the irradiance ratios at the next measured wavelengths. When the criterion is met, the reading at the lowest wavelength is considered the start irradiance level. In case one of the irradiance readings at lower wavelengths is higher, the start irradiance level is increased to the highest irradiance level obtained in the noisy spectral region. This higher value is taken as an indication that the noise level is higher than the level obtained following the ratio method. Two numeric values are output in the detailed flag description of the spectrum: irradiance at the first reliable reading, and the highest irradiance below the first reliable reading. The colour of the flag is determined on the basis of the following criteria set for the higher of the two irradiance values: GREEN < 5e-4 W m-2 nm-1 < YELLOW < 1.5e-3 W m-2 nm-1 < RED < 5e-3 W m-2 nm-1 < BLACK. If the median irradiance level around 310 nm is lower than 5e-4 W m-2 nm-1, the flag is set as GREY."

In addition, we have added the following information in the general description of the ShicRIVM program:

"The method used to determine the lowest detectable irradiance level has been applied to several different instruments with readings at a wide range of solar zenith angles and compared to the method developed by Bernhard et al., 1998, applying the ratio of subsequent spectral measurements from a single instrument. The results were comparable."

We have also added the following reference in the list of references:

Bernhard, G., Seckmeyer, G., McKenzie, R., & Johnston, P. (1998). Ratio spectra as a quality control tool for solar spectral UV measurements. Journal of Geophysical Research: Atmospheres, 103(D22), 28855-28861.

Q: P6 Spike_shape: again a clearer description is needed e.g. (line 24) "the spectral irradiance reading at the measured wavelength and the median of 10 readings around the measured wavelength is over twice that of the matching model calculation: : :." As with Start-irr, the model should be explained somewhere before this point. Is it from

AtmosphericSignature, or is it from within shicRIVM (as implied by Figure 1)? Line 30 "subsequent readings" could mean two scans one after the other. What I think you mean is two consecutive wavelengths in a single spectrum (measured or modeled). Please clarify. See also comment on Start_irr, which I think suffers from the same confusion.

A: We have added a short description of the model to the general description of Shi-cRIVM. The model is from ShicRIVM. Please see our answer to Q: P6 Shift1: above.

We agree with the clarification suggested by the Referee. The assumptions made by the Referee are correct: two readings at two consecutive wavelengths in a single spectrum are used. ShicRIVM is designed to only use single spectra and instrument and location characteristics (such as slit functions, location).

C: We have followed the suggestion of the Referee and replaced the expression "two subsequent readings" by: "two irradiance readings at consecutive wavelengths in the spectrum".

Q: P7 Scan-Variability_2: Please describe fully (see above comment on Shift2). Figure 1 – this does not entirely agree with Table 6, nor with the description of the master flag in the text. The master flag in Figure 1 is stated as dependent on (taken as the worst of) wavelength errors, spectral shape errors (ie spikes) and irradiance scale errors (start irradiance). It does not include (according to Figure 1) the 2 versions of atmospheric transmission flag (from shicRIVM and AtmosphericSignature) that are included in table 6 as contributing to the master flag. Nor does it address the scan variability flag that has been ignored for the Brewer. Figure 1 is operational at the database and implies that a user can select a master flag that for the most part indicates instrument based quality, and then one or both (?) of two atmospheric condition indicators (that are not identical but very similar in their information). The manuscript should explain what a general user of the database can expect from the quality flags (as per Figure 1). If the Sodankyla data has combined the instrument master flag with the atmospheric flags

to give an overall flag then that should be explained separately. If not, then the page on the database for the user interface needs to be changed to be consistent with the applied meaning of master flag.

A: The current version of CheckUVSpec implemented in the database indeed uses the indicator atm_signature in addition to the indicators shift1, shif2, start_irr, spike_shape to determine the master flag. The Referee is therefore pointing out an important deficiency in the description of the master flag in the interface of the database, on the sub page shown in Fig. 1 in the manuscript. The description given on the page states that "Quality flag refers to the master flag for wavelength scale errors, spectral shape errors and irradiance scale errors." This gives an impression that the colours received by the indicators shift1, shif2, start_irr, spike_shape determine the colour of the master flag, which is not the case. The description in the database interface has to be corrected to comply with the current algorithm.

For Brewer #037, the core indicators determining the colour of the master flag are atm_signature, shift1, start_irr, and spike_shape, since the wavelength scale of the instrument does not extend to the wavelength range on which shift2 operates. The worst value of any of these indicators determine the colour of the master flag. If one of them is YELLOW and all others are green, the master flag becomes YELLOW. If one of them is RED and all others are GREEN or YELLOW, the master flag becomes RED. If one of them is BLACK and all others are GREEN, YELLOW, or RED, the master flag becomes BLACK. In case any of these indicators is GREY, the master flag is also set as GREY, independent of the other colours received by the other indicators.

C: To clarify the determination of the master flag, we have added the following paragraph in Materials and methods after the description on ShicRIVM:

"Master flag

The indicators determining the colour of the Master flag are shift1, shift2, start_irr, spike_shape, and atm_signature. In case the wavelength scale of the instrument does

not extend to the wavelength range on which shift2 operates, the indicator shift2 is not taken into account in the determination of the Master flag. The worst value of any of these indicators determine the colour of the master flag. If one of them is YELLOW and all others are green, the master flag is YELLOW. If one of them is RED and all others are GREEN or YELLOW, the master flag is RED. If one of them is BLACK and all others are GREEN, YELLOW, or RED, the master flag is BLACK. In case any of these indicators is GREY, the Master flag is also set as GREY, independent of the other colours received by the other indicators."

This description replaces the following few sentences now deleted just before the description on AtmosphericSignature:

"The master flag is determined by the worst flag for any of the quality indicators. For a GREEN master flag, all indicators have to be flagged as GREEN. In case any of the indicators is BLACK, the master flag is BLACK."

We have also added a note in the description of the quality indicator Shift2 explaining that the GREY value given to spectra not extending over the investigated wavelengths does not affect the master flag. The sentence reads as follows:

"In case the wavelength scale of the instrument does not extend to the wavelength range on which shift2 operates, the indicator shift2 is set as GREY to all spectra and is not taken into account in the determination of the Master flag."

Q: P8, line 19 .. the annual total number of spectra: : : Line 22 – why is there so little data in 2011. If this aspect of the manuscript now indicates the QA of the Sodankyla data we should be told.

A: The reason for the small amount of spectra was found only after the submission of the manuscript. There had been a discontinuity in the submission procedure and the data had been accidentally left non-submitted. The data were uploaded into the database and the statistics were re-calculated. After the submission of the missing

2011 spectra, the total number of 2011 spectra increased from 876 to 5418. This can be now also seen in Fig. 2 showing the monthly amounts of spectra n the database.

C: We have revised the first paragraph in chapter "Results" to comply with the current availability of data in the database. The original text:

"The statistical calculations performed with the Perl script on the flag data retrieved for the Sodankylä Brewer #037 from the EUVDB were examined in detail and summarized. According to the calculations, the total number of spectra was found to vary between 4656 and 6724, except for the year 2011 for which only 876 spectra were found in the database. The same phenomenon could be seen in the tabular and graphical summary provided by the database interface. In Fig. 2, the monthly amounts of spectra over the year 2011 differ from those of the other years."

-> now reads as follows:

"The statistical calculations performed with the Perl script on the flag data retrieved for the Sodankylä Brewer #037 from the EUVDB were examined in detail and summarized. The measurements of solar UV irradiance using Brewer #037 were started on the 5th of April 1990. The total number of spectra until the end of 2014 is 133444. For the year 1990, the database includes 2519 UV irradiance spectra. Over the years 1991-2014, the total annual number of spectra varies between 4656 and 6724, the average annual amount of spectra being 5455."

Q: P9 line 3 Suggest "This is a frequent occurrence for Sodankyla, located within the Polar circle, where the sun can be low for several consecutive scans after sunrise and before sunset."

A: We agree with the Referee and follow this suggestion.

C: We have replaced the sentence on P9 line 3 with the one suggested by the Referee as follows:

"This is a frequently occurring situation for Sodankylä locating beyond the polar circle

where the Sun might be low over several consequtive scans after the sunrise and before the sunset." -> "This is a frequent occurrence for Sodankyla, located within the Polar circle, where the sun can be low for several consecutive scans after sunrise and before sunset."

Q: [Page 9] Lines 7-17 Rather clumsily written. See also comment on figure 1 and develop the argument (eg should Atm_signature be part of the master flag?) The combination of figures 3-7 should be explored. Figs 3 and 4 can definitely be combined, indeed are more instructive that way. Figure 5 might also be added. Alternatively Figure 5 could be combined with Figs 6&7. The case studies are useful. It would also be helpful to show how selecting a certain flag would alter the data set eg select only master flag green and show how that influences the entire Sodankyla dataset – contrast to Fig 2.

A: We agree with the referee that the graphs can be combined. We also modified the script used to calculate the statistics on the flag colours and derived the monthly distributions of the colours for each quality indicator. We prepared a bar chart on the results obtained for the master flag and included it in the manuscript.

C: We have rephrased the paragraph on Page 9 on Lines 7-17 to read as follows:

"According to the results listed in Table 5, the master flag given to the spectra is GREEN in only about 61 % of the cases. The relatively large fraction of the GREY flags received by the indicator Atm_signature (23%) is the major reason for the low amount of GREEN master flags. Part of the GREY flags may be traced to the cases with high solar zenith angle that Sodankylä as a high latitude site has got plenty of. Cases with high solar zenith angle are challenging to both the spectrometer measuring solar UV irradiance and to the model simulating solar UV irradiance to be used as a reference by the QA tool. From the perspective of QA, more emphasis should be therefore put onto the other indicators. However, the Atm_signature flag should yield useful information on the prevailing atmospheric conditions. For Sodankylä Brewer #037, the Atm_signature

could be therefore used to extract cases representing particular measurement conditions of interest."

We have also included a bar chart on the monthly distribution of the colours received by the master flag (as Figure 3) and a paragraph (in Results) on the features seen in the distribution. The paragraph reads as follows:

"Figure 3. presents the monthly distribution of the colours received by the master flag. Ss may be expected, the share of the GREEN flags is smaller in the winter than in the summer. In December, all master flags are GREY. The share of the GREEN flags is at largest in June (77 %). The share of the BLACK flags is at largest in July (0.23 %)."

Following the suggestion of the Referee, we have combined Figs. 3-5 (now as Fig. 4) and Figs 6-7 (now as Fig. 5).

Q: P12 End of conclusion. The work done here has been performed and presented by those very familiar with the EUVDB and the Sodankyla Brewer in its unique setting. The last paragraph of the conclusion states that the master flag is not the most relevant overall, and more detailed exploration of flags (presumably aided by prior knowledge) is required. How would a novice user fare when trying to use the site and QA system. Could a comment on this be provided.

A: We can see the need for a comment on this and we have now addressed this issue in Conclusions.

C: We have added the following two sentences at the end of the last paragraph of Conclusions (P12 line 3):

"The results of this study support the view that the user of the database should familiarize himself/herself with the relevant documentation on the flagging system and the detailed flag information to be able to fully exploit the system. In addition, cooperation with the data provider, who has got the best knowledge of the data, is highly recommendable to any user examining any data retrieved from the database."

Minor points:

Q: P3 Line 4 What is "the planned study". Better just to say "according to the requirements of the user".

A: We agree with the Referee. Indeed this expression is better.

C: The sentence has been rephrased according to the suggestion made by the Referee as follows:

"In addition, the tools are meant to enable selection of data according to the requirements set by the objectives of the planned study."

-> "In addition, the tools are meant to enable selection of data according to the requirements of the user."

Q: P4 line 7 aspects of what?

A: Indeed, the end of the sentence is lacking a few words. We have now added these words.

C: The sentence now reads:

"Inclusion of several different indicators allows checking of the data for different aspects of quality and potential atmospheric conditions."

Q: P4 line 27 grammar Multiple cases of misuse of prepositions. These do not detract from the meaning but should be corrected in editing (one example contributes to the problems in line 27 above).

A: Thanks to this comment, we noticed that there are multiple cases with a preposition "on" occurring with the word "description" in the manuscript. We also noted that there should be no preposition "as" used with the verbs "denote" and "name".

C: We have checked the prepositions and made the following corrections:

P4 line 27: "Description on" -> "Descriptions of"

P6 line 1: "Descriptions of" -> "Descriptions of"

P7 line 6: "Descriptions for" -> "Descriptions of"

P7 line 11: "descriptions on" -> "descriptions of"

Caption of Table 1: "descriptions for" -> "descriptions of"

Page 4 line 9: "denoted as" -> "denoted"

P4 line 27: "denoted as" -> "denoted"

P6 line 1: "named as" -> "named"

Q: P10, Case study 3, and first paragraph of conclusion on P11 – rewrite (clumsy construction)

A: We realize that the description in case study 3 and the first paragraph of conclusions needs clarification. We have followed the suggestion of the Referee.

C: The description of the Case 3 is rephrased as follows:

"This spectrum represents a case where the first reliable irradiance reading in the scan is encountered at a wavelength too far in the scale. The algorithm has distinguished the first reliable reading 0.034 W/m2/nm as far as at 319.5 nm." -> "The spectrum is an example of a case where the algorithm detects the first irradiance reading from the noise as far as at 319.5 nm at the level of 0.034 Wm-2nm-1. The reading exceeds the limit set for the BLACK flag (5e-3 Wm-2nm-1) and hence the flag is set BLACK."

We have also rephrased the first paragraph in Conclusions as follows:

"Solar spectral UV irradiance data measured in Sodankylä by Brewer #037 spectrora-diometer over the years 1990-2014 were studied through the repository features and the QA tools provided by the European UV Database (EUVDB). The summaries on the data give an overview on a consistent dataset extending over quarter of a century with only minor gaps. The gaps found in the time series could be primarily traced to

lamp measurements required for the maintenance of calibration, and intercomparison campaigns that the instrument had been participating, thus not being in operation by the home site at the time." -> "The quality of solar spectral UV irradiance measured by Brewer #037 spectroradiometer in 1990-2014 in Sodankylä, Finland, was examined using the quality assurance (QA) tools provided by the European UV DataBase (EUVDB). Data on the quality indicators, determined by the QA tool and attached to the spectra stored in the database, were retrieved and analyzed. The data set appeared to extend over a quarte of a century with only minor gaps. The gaps in the time series could be traced to annual maintenances and total column ozone calibrations, intercomparison campaigns in Finland and abroad, regular lamp measurements in the laboratory for UV irradiance calibration, and occasional malfunctions due to, e.g., problems with the software."
* * *
**bre037**

**Fig. 1.** Monthly number of spectra measured by Brewer #038 spectrophotometer in Sodankylä
in 1990-2014 and submitted into the European UV Database (EUVDB)

**Fig. 2.** Monthly distribution of colours received by the quality indicators used by the QA tool in the EUVDB.

[Figure]

Fig. 3. Case spectra 1-3 (all GREEN, Shape_spike BLACK, Start_irr BLACK)

[Figure]

**Fig. 4.** Case spectra 4-5 (Shift1 GREY, Transmission_2 BLACK)

---

## Author Comment (AC2) · 17 Jun 2016

Answers from the authors to the Interactive comment on "European UV DataBase (EU-VDB) as a repository and quality analyzer for solar spectral UV irradiance monitored in Sodankylä" by A. Heikkilä et al.

Anonymous Referee #2

The comments are answered below in the following sequential manner: Q denoting the original comment; A denoting the authors' answer to the comment, and C denoting the corrections and amendments to the manuscript.

[Figure]

The authors wish to thank the Referee for his/her invaluable comments and suggestions that assisted in improving the manuscript. The authors also highly appreciate the suggestion on a follow-up study and aim at realizing and reporting on such a study.

General comment: Heikkilä et al., "European UV Database as a repository and quality analyzer for solar spectral UV irradiance monitored in Sodankylä" The authors describe the quality assurance (QA) methodology that is currently used with the solar spectral irradiance measurements. Their approach comprises several metrics that provide important supplemental information about the actual spectral data. This metadata, or, in the authors' terminology, "flags", allows the end-users to assess the reliability of the data. For those actually carrying out the measurements using a spectroradiometer, the QA is an invaluable tool for instrument maintenance and calibration,which is crucial for research based on data covering several decades. In my opinion, the manuscript is of relevance for the science community and suitable for publication in Geoscientific Instrumentation, Methods and Data Systems. I do, however, have a few comments and recommend a minor revision before publication.

Q1. The definition of high quality is discussed in the introduction but only references to literature (Webb et al. and Seckmeyer et al.) are provided. In my opinion, the manuscript would benefit from having a short qualitative description of what actually is considered "standard quality".

A1: We realize that this is likely a question of interest to the readers. Even though no actual standard has been developed, the scientific community does pursue high quality data by following the guidelines jointly prepared. These guidelines are included in the referenced literature. They comprise of lists of specifications for the instruments in use that must be fulfilled simultaneously, several methods for instrument characterizations and maintenance and - last but not least - a number of careful quality checks which should be performed by the operator. Unfortunately, describing the guidelines in full detail is beyond the scope of this paper. However, we have added a note on this issue in the chapter of Results and discussion.

C1: We have added the following paragraphs into the chapter "Results and discussion" and we have included the references therein in the References of the manuscript:

"The quality of solar spectral UV irradiance measurements has been addressed and exhaustively discussed ever since the launch of the first long-term monitoring programs in the late 1980's. While there is no actual standard up to date defining the requirements set to high quality solar UV irradiance data, a common understanding on the requirements is shared by the scientific community and documented in the reports prepared by international advisory groups (Webb et al. 1998, 2003; Seckmeyer et al. 2001, 2005, 2010).

In general, the required quality depends on the scientific question. These could be site specific issues or questions in a wider context, analyzing geographical differences and their causes, for example, as has been done by Seckmeyer et al. (2008a, 2008b). For these two studies, spectra with green flags have been used only. Alternatively, the analysis may focus on a specific question like estimating probability functions (Voskrebenzev et al, 2015), where more spectra with non-green flags may be included."

References:

Seckmeyer, G., Glandorf, M., Wichers, C., McKenzie, R., Henriques, D., Carvalho, F., Webb, A., Siani, A.-M., Bais, A., Kjeldstad, B., Brogniez, C., Werle, P., Koskela, T., Lakkala, K., Gröbner, J., Slaper, H., den Outer, P., & Feister, U. (2008a). Europe's darker atmosphere in the UV-B. Photochemical & Photobiological Sciences, 7(8), 925-930.

Seckmeyer, G., Pissulla, D., Glandorf, M., Henriques, D., Johnsen, B., Webb, A., Siani, A.-M., Bais, A., Kjeldstad, B., Brogniez, C., Lenoble, J., Gardiner, B., Kirsch, P., Koskela, T., Kaurola, J., Uhlmann, B., Slaper, H., den Outer, P., Janouch, M., Werle, P., Gröbner, J., Mayer, B., de la Casiniere, A., Simic, S., & Carvalho, F. (2008b). Variability of UV irradiance in Europe. Photochemistry and Photobiology, 84(1), 172-179.

Voskrebenzev, A., Riechelmann, S., Bais, A., Slaper, H., & Seckmeyer, G. (2015). Estimating probability distributions of solar irradiance. Theoretical and Applied Climatology, 119(3-4), 465-479.

Q2. Likewise, a brief description of the Brewer and its nominal operating mode(s) would be good to have. Perhaps the authors could also describe some of the routine operation challenges, if any, that can or could be effectively tackled by using the QA system rather than on-site routines.

A2: We agree with the Referee as we realize that this could be of interest to the readers.

C2: We have added a brief general description on the nominal operating modes of Brewer, reading as follows:

The Brewer spectrophotometer is primarily used to measure atmospheric total column ozone and solar spectral UV irradiance (Bais et al., 1996; Brewer, 1973). In addition, its measurements may be used to derive atmospheric sulphur dioxide $SO_2$ (Cappellani and Bielli, 1995), nitrogen dioxide $NO_2$ (e.g. Cede et al. 2006; Diémoz et al. 2014 ), and aerosol optical depth (Gröbner et al. 2001; Kazadzis et al. 2005; Marenco et al. 2002). The instrument consists of foreoptics to collect photons of solar UV radiation, a monochromator to separate the irradiance (photons) into spectral components at specific wavelengths, a photomultiplier tube as a radiation detector, and a sun tracker to follow the position of the Sun in the sky. Brewer#037 MkII spectrophotometer in Sodankylä employs a single monochromator, Rejection of stray light is more challenging to the single than to double monochromators, especially at wavelengths below 305 nm (Bais et al., 1996). The wavelength range of the instrument in 290-325 nm.

The Brewer spectrophotometer is designed to operate fully automatically following a schedule predefined by the operator. The schedule contains command strings, each meaning a measurement or an instrumental test performed by the Brewer. Measurement of solar UV irradiance spectrum is scheduled to take place at least every half an hour, typically every twenty minutes. Measurements for total column ozone are

done between the UV scans, either as direct sun, zenith sky or focused sun measurements, depending on the air mass (Karppinen et al., 2016). The schedules have slightly changed over time but the main principles have stayed as described above. Currently, the schedule is defined for each day separately to optimize the number of measurements. Between the sky measurements, the spectrometer makes tests, which are used as quality control (QC) tools to monitor, for instance, the performance of the motors aligning the optics and the photomultiplier tube detecting the photons.

References:

Bais, A., Zerefos, C. and McElroy, C.: Solar UVB measurements with the double- and single-monochromator Brewer Ozone Spectrophotometers, Geophys. Res. Lett., 23, 833–836, 1996.

Brewer, A. W.: A replacement for the Dobson spectrophotometer?. Pure Appl. Geophys., 106-108, 919–927, 1973.

Cappellani, F. and Bielli, A.: Correlation between SO2 and NO2 measured in an atmospheric column by a Brewer spectrophotometer and at ground-level by photochemical techniques, Environmental Monitoring and Assessment, Vol 35, 2, 77-84, 1995. Cede, A., J. Herman, A. Richter, N. Krotkov, and Burrows, J.: Measurements of nitrogen dioxide total column amounts using a Brewer double spectrophotometer in direct Sun mode, J. Geophys. Res. 111, D05304, doi:10.1029/2005JD006585, 2006.

Diémoz, H., Siani, A. M., Redondas, A., Savastiouk, V., McElroy, C. T., Navarro-Comas, M., and Hase, F.: Improved retrieval of nitrogen dioxide (NO2) column densities by means of MKIV Brewer spectrophotometers, Atmos. Meas. Tech., 7, 4009-4022, doi:10.5194/amt-7-4009-2014, 2014.

Gröbner, J., R. Vergaz, V. E. Cachorro, D. V. Henriques, K. Lamb, A. Redondas, J. M. Vilaplana, and Rembges, D.: Intercomparison of aerosol optical depth measurements in the UVB using Brewer spectrophotometers and a Li-Cor spectrophotometer,

Geophys. Res. Let. 28, 1691-1694, 2001.

Karppinen, T., Lakkala, K., Karhu, J. M., Heikkinen, P., Kivi, R., and Kyrö, E.: Brewer spectrometer total ozone column measurements in Sodankylä, Geosci. Instrum. Method. Data Syst., 5, 229-239, doi:10.5194/gi-5-229-2016, 2016.

Kazadzis, S., Bais, A., Kouremeti, N., Gerasopoulos, E., Garane K., Blumthaler, M., Schallhart, B. and Cede A.: Direct spectral measurements with a Brewer spectrora-diometer: Absolute calibration and aerosol optical depth retrieval, Appl. Opt., 44(9), 1681 – 1690, 2005.

Marenco, F., A. di Sarra, and De Luisi, J.:Methodology for determining aerosol optical depth from Brewer 300-320-nm ozone measurements, Appl. Opt., 41, 1805-1814, 2002.

In Discussion, we have also inserted the following paragraph dealing with the potential use of the QA tools to meet the operational challenges:

"Currently, the QA tools of the EUVDB are mainly used to complement the on-site QC routines. In addition, they could be used to remotely monitor the performance of the instrument at an unmanned station. If the spectra were automatically uploaded into the EUVDB, the QA flags of the database could alert on a problem with the wavelength setting (Shift1, Shift2), or snow/dirt covering the entrance optics of the instrument and blocking the incoming radiation (Start_irr, Spike_shape, Too low irradiance). They could be also used to separate scans made under changing cloud conditions (Spike_shape, Moving clouds), in case the data is used for validation of near-real time satellite data or model calculations. As one solar UV scan takes up to 3 minutes, the cloud conditions may change during the scan, affecting the reliability of the comparison."

Q3. (Results and discussion) Does the number of spectra (4656-6724) refer to the annual measurements? Why does this vary? Instrument trouble or do you only carry

out measurements when certain criteria are met?

A3: Yes, the number of spectra refer to the annual number of scans of solar spectral UV irradiance. The annual amount of scans vary due to several reasons, including instrument trouble. Five main factors affecting the annual number of collected scans may be distinguished:

1. Brewer #037 has been calibrated for total column ozone measurements according to the list given in the table below (published in Karppinen et al 2016). Some of the calibrations have been performed at the home site of the instrument (Sodankylä) whereas some of them have been realized at other sites within measurement intercomparison campaigns. Calibration performed in Finland (in Sodankylä or in Jokioinen) has caused a break of 5-7 days into the time series of the solar UV scans. The gap caused by an intercomparison campaign is longer, typically from 2 weeks to 1 month. The lengths and timings of these gaps vary from year to year, resulting in variability in the amount of solar UV scans collected annually at the home site of Brewer #037.

1988 April Sodankylä 1989 June Sodankylä 1990 June Sodankylä 1993 November Izaña 1994 September Jokioinen 1995 June Sodankylä 1996 October Izaña 1997 July Sodankylä 1998 June Jokioinen 1999 June Jokioinen 2000 June Tylosand 2001 June Jokioinen 2002 June Sodankylä 2003 June Sodankylä 2004 June Jokioinen 2005 June Sodankylä 2006 June Jokioinen 2007 May Sodankylä 2008 June Jokioinen 2009 June Sodankylä 2009 December Izaña 2011 November Izaña 2013 November Izaña

2. Brewer #037 has been calibrated for UV irradiance by performing lamp measurements in the optical laboratory of the Arctic Research Center of the Finnish Meteorological Institute in Sodankylä. Typically, the frequency of these measurements has been 6-8 weeks. However, there are year-to-year differences in the frequency due to, for instance, the availability of personnel capable of performing the measurements.

3. The operating software of Brewer #037 seizes up from time to time. This may have resulted in a loss of several UV scans, depending on how quickly the operator has

noticed the jam. The software has been under long-term development by the supplier IOS Inc. over the years, resulting in a number of updated versions of the software with enhanced operational reliability. Some of the versions have been more prone to seize up than the others. Hence, the number of jams due to the software varies from year to year. Recently, this issue has been addressed by incorporating the measurements into the operative 24/7 control system of the FMI observational services. The system alerts immediately in case of malfunctions so that the measurements can be restarted and no large gaps are formed in the daily data set (Mäkelä et al. 2016).

4. The frequency the Brewer #037 performs solar UV scans is regulated by the pre-defined schedules. The schedules have been updated over the operational years. Unfortunately, there has been no system to keep track on the changes made in the schedules. Improved sampling of the diurnal cycle of the solar UV irradiance has been one of the objectives when redefining the schedules. The number of scheduled daily UV scans has therefore likely increased over the years. This could be verified by examining the days with uninterrupted sky measurements for the daily number of scans. While an exhaustive analysis would have been beyond the scope of this study, we selected two pairs of uninterrupted measurement days in June and July in 1991 and 2014. The result was as follows:

16891 17 Jun 1991 Number of spectra: 24 (first: 00:26:31 UTC; last: 20:22:36 UTC)
16814 16 Jun 2014 Number of spectra: 31 (first: 00:19:24 UTC; last: 21:42:20 UTC)

19891 17 Jul 1991 Number of spectra: 23 (first: 00:38:53 UTC; last: 19:44:39 UTC)
19814 16 Jul 2014 Number of spectra: 31 (first: 00:35:10 UTC; last: 20:40:48 UTC)

The daily amount of UV scans is indeed larger in 2014 than in 1991.

5. The QC/QA procedures reject part of the measured spectra as erroneous. All data submitted to EUVDB is subject to final (Level 2) QA including wavelength correction employing the program ShicRIVM. All the scanned spectra are also visually inspected and compared against ancillary broadband and modelled UV data (Lakkala

et al. 2008), and clearly erroneous spectra are rejected. Typically five to ten spectra are rejected at that stage. This QA procedure has been followed since 2005. The minimum number of annually rejected spectra per year since 2005 is one (in 2010 and in 2014). The corresponding maximum number of spectra is 16 (in 2007). Over the years 1990-2004, the number of spectra rejected at the final stage of QA has been larger. Especially during the first few operational years, there were problems with the data transfer, for instance, resulting in occasional corruption of transferred data files, which increased the number of rejected scans.

All of the above mentioned factors introduce variability to the annual number of collected scans of solar UV irradiance. We estimate that the first three factors are more significant than the last two. The bar chart below shows the development of the annual number of spectra. The inter-annual variability is large, but the number appears to have grown from the 1990's (please see the supplementary Fig. 1).

We have rewritten the first paragraph of Chapter 3 and included a brief explanation to the variability of annual amounts of scans, as we can see that this might be of interest to the readers.

References:

Karppinen, T., Lakkala, K., Karhu, J. M., Heikkinen, P., Kivi, R. & Kyrö, E. (2016). Brewer spectrometer total ozone column measurements in Sodankylä. Geoscientific Instrumentation, Methods and Data Systems Discussions, 2016, 1-18. doi:10.5194/gi-2015-41

Lakkala, K., Arola, A., Heikkilä, A., Kaurola, J., Koskela, T., Kyrö, E., Lindfors, A., Meinander, O., Tanskanen, A., Gröbner, J. & Hülsen, G. (2008). Quality assurance of the Brewer spectral UV measurements in Finland. Atmospheric Chemistry and Physics, 8(13), 3369-3383.

Slaper, H., Reinen, H., Blumthaler, M., Huber, M. & Kuik, F. (1995). Comparing ground‐level spectrally resolved solar UV measurements using various instru-
ments: A technique resolving effects of wavelength shift and slit width. Geophysical
Research Letters, vol. 22, no. 20, pp. 2721-2724.

C3: We have added the following text in the beginning of the Chapter 3:

The annual amount of scans vary due to several reasons. Five main factors affecting
the annual number of collected scans may be distinguished, described briefly in the
following:

"1. Annual maintenance and calibrations for total column ozone measurements have
caused breaks of varying durations in the solar UV measurements. Calibration per-
formed in Finland (in Sodankylä or in Jokioinen) has caused a break of 5-7 days into
the time series of the solar UV scans. The gap caused by an intercomparison cam-
paign abroad has been longer, typically from 2 weeks to 1 month. The lengths and
timings of these gaps vary from year to year, resulting in variability in the amount of
solar UV scans collected annually at the home site of Brewer #037.

2. Brewer #037 has been calibrated for UV irradiance by performing lamp measure-
ments in the optical laboratory of the Arctic Research Center of the Finnish Meteo-
rological Institute in Sodankylä. Typically, the frequency of these measurements has
been 6-8 weeks. However, there are year-to-year differences in the frequency due to,
for instance, the availability of personnel capable of performing the measurements.

3. The operating software of Brewer #037 seizes up from time to time. This may
have resulted in a loss of several UV scans. The software has been under long-term
development by the supplier IOS Inc. over the years, resulting in a number of updated
versions of the software with enhanced operational reliability. Some of the versions
have been more prone to seize up than the others. Hence, the number of jams due to
the software varies from year to year.

4. The frequency the Brewer #037 performs solar UV scans is regulated by pre-defined

schedules. The schedules have been updated over the operational years. Improved sampling of the diurnal cycle of the solar UV irradiance has been one of the objectives when redefining the schedules. The number of scheduled daily UV scans has therefore increased over the years.

5. The on-site QC/QA procedures reject part of the measured spectra as erroneous. All data submitted to EUVDB is subject to final (Level 2) QA including wavelength correction employing the program ShicRIVM. All the scanned spectra are also visually inspected and compared against ancillary broadband and modelled UV data (Lakkala et al. 2008), and clearly erroneous spectra are rejected. Typically five to ten spectra are rejected annually at that stage. This QA procedure has been followed since 2005. The minimum number of annually rejected spectra per year since 2005 is one (in 2010 and in 2014). The corresponding maximum number of spectra is 16 (in 2007). Over the years 1990-2004, the number of spectra rejected at the final stage of QA has been larger.

All of the above mentioned factors introduce variability to the annual number of collected scans of solar UV irradiance. The first three factors may be estimated more significant than the last two."

We have also added the following references in the list of references:

Karppinen, T., Lakkala, K., Karhu, J. M., Heikkinen, P., Kivi, R. & Kyrö, E. (2016). Brewer spectrometer total ozone column measurements in Sodankylä. Geoscientific Instrumentation, Methods and Data Systems Discussions, 2016, 1-18. doi:10.5194/gi-2015-41

Lakkala, K., Arola, A., Heikkilä, A., Kaurola, J., Koskela, T., Kyrö, E., Lindfors, A., Meinander, O., Tanskanen, A., Gröbner, J. & Hülsen, G. (2008). Quality assurance of the Brewer spectral UV measurements in Finland. Atmospheric Chemistry and Physics, 8(13), 3369-3383.

[Figure]

Slaper, H., Reinen, H., Blumthaler, M., Huber, M. & Kuik, F. (1995). Comparing ground‐level spectrally resolved solar UV measurements using various instruments: A technique resolving effects of wavelength shift and slit width. Geophysical Research Letters, vol. 22, no. 20, pp. 2721-2724.

Q4: (Results and discussion, page 10, lines 22-28) The authors state that a detailed examination of the selected cases provides a more profound understanding of the function and performance of the QA methodology. While I agree that a closer look at the data does help in understanding why a certain flag is there, I don't think a small number of cases is sufficient for generalisation. Are you really sure that you would have arrived to the same conclusions if you had selected different spectra? Wouldn't it be much more useful to collect all spectra with, e.g., Shift1 GREY flag and analyse why the algorithm (built-in to the QA) cannot make any conclusions about wavelength scale shifts? Something like this would be an excellent topic for a follow-up study.

A4: We agree with the Referee on his/her view that the case study presented here cannot result in a comprehensive analysis on the performance of the QA tools. For a deliverable of this kind, the study should exhaustively include all the spectra in the database. Alternatively, a representative sample of spectra could be used. Indeed, retrieval and investigation of all spectra flagged as GREY for a particular quality indicator, like Shift1 targeted to detect the shifts in the wavelength scale, would be extremely interesting. We highly appreciate this suggestion and will certainly aim at a follow-up study on the topic.

C4: The sentences on lines 22-28 in Chapter 3 (Results and discussion) has been rephrased to make the scope of the study more clear as follows:

"Analysis on the statistics of the flag information is obviously an efficient way to get an overall view on different aspects of the data quality. However, the detailed examination of the selected cases as described above gives a more profound insight into the function and performance of the QA tools implemented in the database. Specifically, an

understanding on the metrics and categorization used by the different quality indicators helps the data provider and the user in analysing and using the data in a meaningful way. Clearly, the indicators provide an added value to the data set." -> "Analysis on the mere statistics of the flag information is obviously an efficient way to get an overall view on different aspects of the quality of the data of interest. However, a detailed examination of selected cases as described above is apt to give an even more profound insight into the data studied and the special characteristics therein. Specifically, an understanding on the metrics and categorization used by the different quality indicators helps the data provider and the user in analysing and using the data in a meaningful way. Clearly, the indicators provide an added value to the data set."

We have also added the following paragraph in the end of Conclusions, to further clarify the scope and limitations of the study:

"The analysis on the performance of the QA tools and the conclusion drawn in this study are strictly valid only for the particular data set studied, i.e., solar spectral UV irradiance measured by Brewer #037 in Sodankylä over the years 1990-2014. Further studies on the performance of the QA tools of the EUVDB should therefore cover a number of measurement sites and instruments. A follow-up study still focusing on the Sodankylä Brewer #037 UV data in its unique setting at a high latitude site is also planned. The study is intended to focus on the GREY flags for each quality indicator separately, to investigate the performance of the algorithm in these undetermined cases exclusively. Compatibility of cloudiness conditions determined by the QA tool and synoptic cloud observations would be another interesting topic for a further study."

Q5. (Conclusions) Are the gaps in the time series not recorded in the EUVDB? Would it not be extremely useful for the end-users to quickly find out that there are no spectra for the time they are interested in?

A5: The time gaps as such are not recorded in the database. However, large gaps may be detected in the graphical and tabular presentations on the monthly amounts

of scans on a subpage of "Site list" giving site specific information on each station. The presentations on the page are based on PL/SQL tools operating into the Oracle database. The data retrieved for the availability of data by using these tools could be also used to compile information on the lack of data. This might be indeed a convenient feature in the database. Alternatively, this information could be collected and made available in a form of a simple list on the time periods with no data, supplemented with an explanation for the lack of data.

C5: We have added a paragraph dealing with the gaps in the time series and the annual variability therein in Chapter 3 (Results and discussion). In addition, Chapter 2 now includes a paragraph describing the tabular and graphical summaries on the monthly amounts of data submitted into the database. In this context, it is now also mentioned that the gaps may be inferred from the summaries. The paragraph inserted in Chapter 2 (Materials and methods) in section "QA tools and flagging" reads as follows:

"The EUVDB contains a specific subpage listing all the sites and instruments registered into the database. The page provides site and instrument specific information for the users of the data. In addition, it gives tabular and graphical summaries on the monthly amounts of solar UV spectra submitted to the database. The summaries may be used as indicators on the availability of data. The database user may find the summaries very helpful since they allow quick browsing of the availability of data, prior to actual data retrieval. They can be also used in an inverse manner to infer amounts of missing data, i.e., gaps in the time series."

Q6:. (Conclusions) There were 23% of GREY flags for the overall quality. The authors state that the majority of these indefinite conclusions could be traced to restrictions in the radiative transfer model FastRT that could not handle solar zenith angles above 84 degrees. Are there better models or has your quality flag analysis highlighted a gap in our knowledge? In both cases, these indefinite cases would probably be of high interested for modellers working on radiative transfer at higher latitudes.

A6: The performance of 1-d radiative transfer models may be enhanced by replacing the plane-parallel layers of the atmosphere with a pseudo-spherical model of the atmosphere. This has been also realised in FastRT, improving its performance at high solar zenith angles (sza) and extending the usability of the model up to sza of at least 84 degrees. The earliest versions of libRadtran (the basis of FastRT), in comparison, performed well up to 80 degrees (Mayer et al. 2007).

3-d radiative transfer models like MYSTIC and McArtim (validated by Mayer et al. (2009) and Deutschmann et al. (2011), respectively) are more accurate than capable of simulating solar UV irradiance even up to 91 degrees. Up to date these kinds of models remain too compute-intensive to be run on a server upon an Oracle database.

We agree with the Referee on the point that the cases flagged as GREY in the database should form a highly interesting data set, potentially useful for model development and validation.

References:

Deutschmann, T., Beirle, S., Frieß, U., Grzegorski, M., Kern, C., Kritten, L., Platt, U., Prados-Román, C., Puki, J. & Wagner, T. (2011). The monte carlo atmospheric radiative transfer model McArtim: Introduction and validation of jacobians and 3D features. Journal of Quantitative Spectroscopy and Radiative Transfer, 112(6), 1119-1137.

Mayer, B., Seckmeyer, G., & Kylling, A. (1997). Systematic long‐term comparison of spectral UV measurements and UVSPEC modeling results. Journal of Geophysical Research: Atmospheres, 102(D7), 8755-8767.

Mayer, B., Hoch, S., & Whiteman, C. (2010). Validating the MYSTIC three-dimensional radiative transfer model with observations from the complex topography of arizona's meteor crater. Atmospheric Chemistry and Physics, 10(18), 8685-8696.

C6: The following paragraph and the references therein has been added into the manuscript (in Discussion) to enlighten the performance of FastRT limiting below szas

of 84 degrees and to emphasize the usability of the cases flagged as GREY in model development and validation:

"The cases flagged as GREY in the database represent a highly interesting data set that could benefit model development and validation. The performance of 1-d radiative transfer models may be enhanced by replacing the plane-parallel layers of the atmosphere with a pseudo-spherical model of the atmosphere. This has been also realised in FastRT, improving its performance at high solar zenith angles and extending the usability of the model up to sza of at least 84 degrees. The earliest versions of libRadtran (the basis of FastRT), in comparison, performed well up to 80 degrees (Mayer et al. 2007). 3-d radiative transfer models like MYSTIC and McArtim (validated by Mayer et al. (2009) and Deutschmann et al. (2011), respectively) are more accurate than capable of simulating solar UV irradiance even up to 91 degrees. Up to date, these kinds of models remain too compute-intensive to be run on a server upon an Oracle database. With the ever advancing computer efficiencies, this may not be the case in the future."

We have also added the following references in the list of references:

Deutschmann, T., Beirle, S., Frieß, U., Grzegorski, M., Kern, C., Kritten, L., Platt, U., Prados-Román, C., Puki, J. & Wagner, T. (2011). The monte carlo atmospheric radiative transfer model McArtim: Introduction and validation of jacobians and 3D features. Journal of Quantitative Spectroscopy and Radiative Transfer, 112(6), 1119-1137.

Mayer, B., Seckmeyer, G., & Kylling, A. (1997). Systematic long‐term comparison of spectral UV measurements and UVSPEC modeling results. Journal of Geophysical Research: Atmospheres, 102(D7), 8755-8767.

Mayer, B., Hoch, S., & Whiteman, C. (2010). Validating the MYSTIC three-dimensional radiative transfer model with observations from the complex topography of arizona's meteor crater. Atmospheric Chemistry and Physics, 10(18), 8685-8696.

Q7. (Table 2 and 3) Have you compared the cloudy flag with synoptic observations?

Do they agree?

A7: Synoptic cloud observations (cloudiness in octas) are available for Sodankylä for the time period 1 Jan 1990 – 4 Feb 2008. Estimates on cloudiness given by an AWS (Automatic Weather Station) are also available starting from 4 Feb 2008 until today. The synoptic observations have been done every three hours until Jun 15, 2001. During the time period 16 Jun 2011 – 9 May 2006, the observation for 02:40UTC has not been done due to changes in the manpower at the observatory. Since 10 May 2006, until Sep 1 2006, the observations for the nighttime hours 20:40UTC, 23:40UTC, and 02:40UTC are not available for the weekends. Starting from Sep 2 2006, the nighttime observations are not available for any day of the week. Since 1 Jan 2008, also the synoptic observation for 17:40UTC is missing. The temporal resolution of the AWS data is 10 minutes.

We have not carried out any systematic comparison between synoptic/automatic estimates on cloudiness, but we realize that this would be a very interesting topic for a study. For the time period 1 Jan 1990 - 4 Feb 2008, the temporal resolution of 3 hours does not allow finding a representative estimate for every solar UV irradiance scan. The 10-min data from AWS, however, may provide estimates reasonably representative for all the moments of solar spectral UV measurements. We will definitely aim at looking into the issue in our further studies.

C. We have added the following sentence in the end of Conclusions:

"Compatibility of cloudiness conditions determined by the QA tool and synoptic cloud observations would be another interesting topic for a further study."

– Some minor comments:

Q8: (Abstract, page 1, lines 22-24): The sentence "Spectra scanned by..." is very complex. Could be simplified.

A8: We agree with the Referee on this point. The sentence has been now simplified.

[Figure]

C8: We have simplified the expression and described the contents of the study more precisely by replacing the sentence by two new sentences as follows:

Spectra scanned by the Brewer#037 MkII spectroradiometer in Sodankylä (67.37 °N, 26.63 °E) over the years 1990-2014 and uploaded into the database are examined using the inherent QA tools of the database.

-> We confine the study on the data measured by Brewer#037 MkII spectroradiometer in Sodankylä (67.37 °N, 26.63 °E) in 1990-2014. The quality indicators associated with the UV irradiance spectra uploaded into the database are retrieved from the database and subjected to a statistical analysis.

Q9: (Introduction, page 3, lines 21-22: I do not understand the sentence "The quality indicators are examined for their frequency in general..." Do you refer to "occurrence"?

A9: Indeed, our intension was to examine exactly the occurrence of the different quality indicators.

C9: The sentence has been rephrased as follows:

"The quality indicators are examined for their frequency in general, and for selected case spectra in detail." -> "The quality indicators are examined for their values (i.e.: colors), and the frequency distribution of the colors, denoting different categories of quality, are derived for each indicator. In addition, selected case spectra, representing different categories of quality, are studied in detail."
* * *
Fig. 1. Number of solar UV irradiance spectra measured annually by Brewer #037 in Sodankylä in 1990-2014